# Phosphorylation regulates viral biomolecular condensates to promote infectious progeny production

Nicholas Grams [1,2], Matthew Charman [1,3✉], Edwin Halko[1], Richard Lauman[1], Benjamin A Garcia[4] & Matthew D Weitzman [1,3,5✉]

## Abstract

**Biomolecular condensates (BMCs) play important roles in diverse biological processes. Many viruses form BMCs which have been implicated in various functions critical for the productive infection of host cells. The adenovirus L1-52/55 kilodalton protein (52K) was recently shown to form viral BMCs that coordinate viral genome packaging and capsid assembly. Although critical for packaging, we do not know how viral condensates are regulated during adenovirus infection. Here we show that phosphorylation of serine residues 28 and 75 within the N-terminal intrinsically disordered region of 52K modulates viral condensates in vitro and in cells, promoting liquid-like properties. Furthermore, we demonstrate that phosphorylation of 52K promotes viral genome packaging and the production of infectious progeny particles. Collectively, our findings provide insights into how viral condensate properties are regulated and maintained in a state conducive to their function in viral progeny production. In addition, our findings have implications for antiviral strategies aimed at targeting the regulation of viral BMCs to limit viral multiplication.**

**Keywords** Adenovirus; Viral Packaging; Phosphorylation; Biomolecular Condensates; Protein Intrinsic Disorder
**Subject Categories** Microbiology, Virology & Host Pathogen Interaction; Post-translational Modifications & Proteolysis

## Introduction

Recent advances indicate that many nuclear and cytoplasmic bodies are biomolecular condensates (BMCs) formed by phase transitions (Banani et al, 2017; Brangwynne et al, 2009; Hyman et al, 2014; Shin and Brangwynne, 2017). BMCs are dynamic, membrane-less assemblies that compartmentalize cellular components to orchestrate biological processes (Banani et al, 2017; Hnisz et al, 2017; Holehouse and Pappu, 2018; Hyman et al, 2014; Li et al, 2012; Shin and Brangwynne, 2017). These condensates can range from liquid-like, to those more closely resembling hydrogels, glasses, or viscoelastic solids (Brangwynne et al, 2015; Gomes and Shorter, 2019; Holehouse and Pappu, 2018; Riback et al, 2017; Shin and Brangwynne, 2017; Weber, 2017). The properties of BMCs depend on the nature of the dynamic multivalent interactions that underpin their formation. The loss of dynamic interactions in favor of more stable interactions is linked with the transition from a liquid-like state to a more gel-, glass-, or solid-like state (Brangwynne et al, 2015; Harmon et al, 2017; Holehouse and Pappu, 2018; Lin et al, 2015; Patel et al, 2015; Riback et al, 2017; Uversky, 2019). Understanding these properties and how they change or are maintained is crucial to understanding the role of BMCs in cellular processes. The emerging importance of BMCs as orchestrators of cellular biology has led to the suggestion that many viral subcellular compartments are BMCs formed by phase transitions such as liquid-liquid phase separation (Brocca et al, 2020; Etibor et al, 2021; Geiger et al, 2021; Heinrich et al, 2018; Hidalgo et al, 2021; Sagan and Weber, 2023; Su et al, 2021). In turn, this has led many to question how the associated biophysics and properties of these compartments contribute to productive virus infection (Brocca et al, 2020; Charman and Weitzman, 2020; Etibor et al, 2021; Geiger et al, 2021; Hidalgo et al, 2021; Mishra et al, 2020; Sagan and Weber, 2023; Su et al, 2021).

Human adenovirus (AdV) is a ubiquitous respiratory and gastrointestinal pathogen (Lion, 2014; Lynch et al, 2011; Lynch and Kajon, 2016). During the late stage of AdV infection, the newly replicated viral dsDNA genome is packaged within a proteinaceous icosahedral capsid shell (Ahi and Mittal, 2016; Charman et al, 2019; San Martín, 2012). Packaging of the viral genome and incorporation of all essential viral structural proteins leads to particle maturation, resulting in infectious progeny particles for spread and transmission (Ahi and Mittal, 2016; Mangel and Martín, 2014; Pérez-Berná et al, 2014). We recently showed that nuclear bodies (NBs) that form during the late stage of AdV infection are BMCs (Charman et al, 2023). These NBs organize viral capsid proteins,

[1]Division of Protective Immunity and Division of Cancer Pathobiology, The Children's Hospital of Philadelphia, Philadelphia, PA, USA. [2]Cell & Molecular Biology Graduate Group, University of Pennsylvania Perelman School of Medicine, Philadelphia, PA, USA. [3]Department of Pathology and Laboratory Medicine, University of Pennsylvania Perelman School of Medicine, Philadelphia, PA, USA. [4]Department of Biochemistry and Molecular Biophysics, Washington University School of Medicine, St. Louis, MO, USA. [5]Penn Epigenetics Institute, University of Pennsylvania Perelman School of Medicine, Philadelphia, PA, USA. ✉E-mail: charmanm@chop.edu; weitzmanm@chop.edu

regulating progeny production such that capsid assembly is coordinated with the provision of viral genomes. The formation of viral NBs requires the viral L1-52–55K protein (52K), and more specifically, its N-terminal intrinsically disordered region (IDR). Modifying the behavior of these BMCs by mutation of a proline-rich spacer region at the extreme N-terminal end of 52K protein IDR correlates with loss of viral progeny production (Charman et al, 2023). This suggests that the underlying IDR-mediated interactions must be finely tuned to maintain viral proteins in a state conducive to downstream capsid assembly. These findings led us to question whether the N-terminal IDR of 52K is regulated to ensure that condensate properties remain conducive to viral progeny production.

When compared to the entire human proteome, posttranslational modifications (PTMs) are over-represented within IDRs (Collins et al, 2008; Iakoucheva et al, 2004; Owen and Shewmaker, 2019). Recent observations highlight the role of protein PTMs, including phosphorylation, as modulators of IDR behavior (Bah and Forman-Kay, 2016; Darling and Uversky, 2018; Duan and Walther, 2015; Sridharan et al, 2022). The addition or removal of a phosphate group can alter the multivalent interactions that underpin BMCs (Bah and Forman-Kay, 2016; Banerjee et al, 2016; Duan and Walther, 2015; Flock et al, 2014; Hofweber and Dormann, 2019; Owen and Shewmaker, 2019). Accordingly, phosphorylation may promote, inhibit, or modify BMCs (Aumiller and Keating, 2016; Bah and Forman-Kay, 2016; Banerjee et al, 2016; Hofweber and Dormann, 2019; Owen and Shewmaker, 2019). Previous reports indicate that 52K is a phosphoprotein (Lind et al, 2012, 2013; Lucher et al, 1986; Pérez-Berná et al, 2014; Valdés et al, 2020). However, whether phosphorylation of 52K contributes to its role in viral progeny production is unknown. This led us to question whether the N-terminal IDR of the 52K protein could be regulated by phosphorylation to promote viral progeny production. Here we show that phosphorylation of specific serine residues within the N-terminal IDR of the 52K protein modulates viral BMCs to promote assembly of complete, packaged, infectious progeny particles.

## Results

### The adenovirus 52K protein is phosphorylated

The 52K protein of AdV serotype 2 has previously been reported to be phosphorylated at serine residues 75 (S75) and 360 (S360) (Lind et al, 2012, 2013; Valdés et al, 2020). To investigate phosphorylation of the 52K protein during AdV infection, we infected human embryonic kidney (HEK-293) cells or human lung epithelial (A549) cells with the closely related AdV serotype 5 and analyzed cell lysates by immunoblot analysis. Whole-cell lysates were treated with the non-specific Lambda Protein Phosphatase (λ-PP). Immunoblotting showed that 52K from λ-PP-treated lysates exhibited greater electrophoretic mobility compared to 52K from control-treated lysates (Fig. 1A), indicating that 52K is dephosphorylated by λ-PP. This confirmed that 52K is phosphorylated during infection of both HEK-293 and A549 cells. To investigate the site specificity of 52K phosphorylation during AdV infection, we searched for sites of protein phosphorylation within a whole-cell proteome dataset from AdV serotype 5-infected A549 cells

previously generated by our lab (Dybas et al, 2021). Our analysis consistently identified several phosphorylated sites on the 52K protein, one of which matched the previously reported phosphorylation of S75 (Fig. 1B,C). In addition, three novel serine phosphorylation events were identified at residues S28, S145, and S336 (Fig. 1B,C). Phosphorylation of S28, S75, and S145 was detected at 16-, 24-, and 48-h post infection (hpi) (Fig. EV1), indicating that phosphorylation of these residues is present throughout the late stage of infection. The previously reported S360 modification was not identified in our dataset (Lind et al, 2012, 2013; Valdés et al, 2020).

Our recent findings suggest that the N-terminus of 52K is intrinsically disordered (Charman et al, 2023). To model 52K protein structure and visualize the location of serine phosphorylation events, we entered the primary amino acid sequence of the 52K protein (Uniprot Q6VGV2) along with the identified PTMs into the AlphaFold 2.0 program (Jumper et al, 2021; Ruff and Pappu, 2021). The resulting proposed structure, displayed as a ribbon model with the identified phosphorylated residues as space-filling, shows that S28 and S75 are within the predicted N-terminal IDR (~140 amino acids). The S145 site sits at the junction between the N-terminal IDR and a coiled region, while S336 resides at the C-terminus away from the IDR (Fig. 1D). Given that posttranslational modification of IDRs can regulate BMCs (Bah and Forman-Kay, 2016; Banerjee et al, 2016; Duan and Walther, 2015; Hofweber and Dormann, 2019; Owen and Shewmaker, 2019; Vuzman et al, 2012), we subsequently focused our studies on phosphorylation within the 52K IDR at residues S28 and S75.

Since AdV does not encode its own kinase, the viral 52K protein must be phosphorylated by a host kinase. To validate phosphorylation of the IDR and predict which kinases might be capable of phosphorylating 52K, we used a bioinformatics approach to match the S28 and S75 phospho-sites to known human kinase substrate motifs (Safaei et al, 2011). This in silico kinase screen implicated kinases belonging to the Cyclin-dependent kinase (CDK), Mitogen-activated protein kinase (MAPK), Glycogen synthase kinase (GSK), and CDC-like kinase (CLK) (CMGC) group of proline-directed serine/threonine kinases (Fig. EV1B,C). To follow-up on in silico prediction scores, we assessed the ability of 25 candidate kinases to phosphorylate peptides containing the S28 or S75 sites in vitro, using the ADP-Glo™ methodology (Zegzouti et al, 2010). Among the kinases that phosphorylated peptides containing S28 or S75, 7 of the top-ranked kinases ($> 1 \times 10^6$ relative light units) belonged to the CMGC group (HIPK1, JNK1, DYRK1a, CDK5/p25, ERK1, CDK1/cyclinA1, and p38Δ) (Fig. EV1D). These data suggest that diverse members of CMGC kinase group possess the ability to phosphorylate the 52K N-terminal IDR at residues S28 and S75 in vitro, and that CMGC group kinases generally recognize these sites better than other kinase groups. Of note, the top-ranked CMGC group kinases are reported to localize to the nucleus (Thul et al, 2017; Uhlén et al, 2005), where 52K accumulates and viral NBs form. Intriguingly, AdV infection is already known to stimulate several CMGC group kinases to make host cells suitable for infection, raising the question of whether kinases of this group are co-opted by the virus for additional purposes (Glenewinkel et al, 2016; Keblusek et al, 1999; Kovesdi and Bruder, 1997; Suomalainen et al, 2001; Yomoda et al, 2008). We next explored whether inhibition of cellular kinases by small molecule inhibitors would shed light on the kinase pathways responsible for

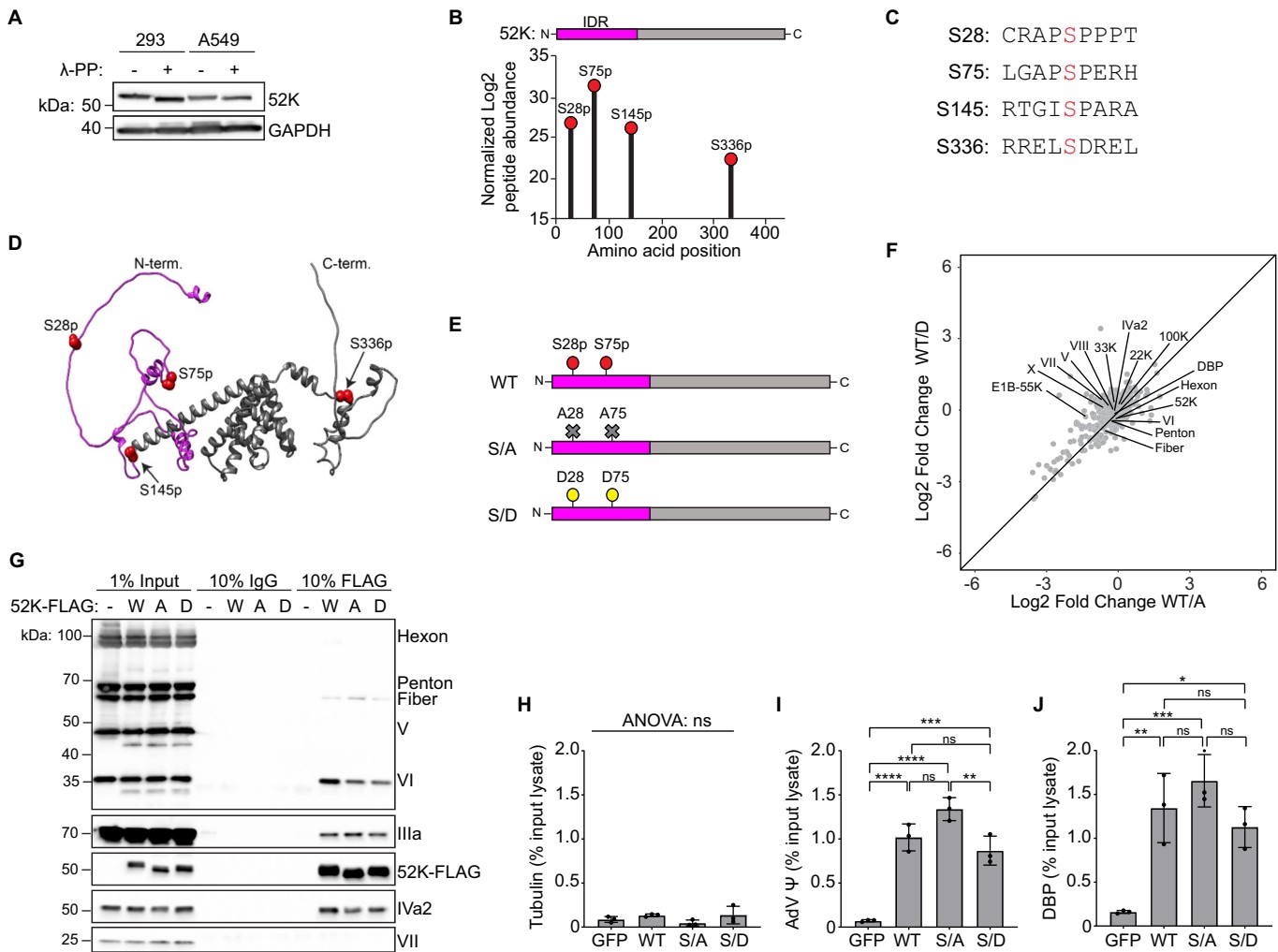

**Figure 1. The AdV 52K protein is phosphorylated during infection.**

(A) Immunoblot showing electrophoretic mobility of the 52K protein, or GAPDH loading control. Lysates from adenovirus (AdV)-infected HEK-293 or A549 cells were treated with (+) or without (−) the non-specific Lambda Protein Phosphatase (λ-PP). Representative of $n = 2$ independent replicates. (B) Site specificity of 52K protein phosphorylation determined by analysis of Post Translational Modification-Mass Spectrometry dataset generated from AdV-infected A549 cells at 24 h post infection. Modifications shown as mean log2 abundance of modified peptide relative to total peptide. A schematic of 52K is shown with the N-terminal Intrinsically Disordered Region (IDR) highlighted (magenta), and the C-terminal region shown in gray. (C) Amino acid sequence surrounding identified phosphorylation sites. Phosphorylated serine residues are shown in red. (D) Alpha-fold 2.0 predicted ribbon structure of the 52K protein with phospho-sites included (red, space-filling). (E) Schematic of wild-type 52K (WT) and phosphorylation-negative S28/75A (S/A) and phosphorylation-mimetic S28/75D (S/D) mutants. Phosphorylated serines are depicted by red circles, alanine mutations are depicted by gray crosses, aspartate mutations are depicted as yellow circles. (F) Cross-plot comparing mean fold change in abundance of proteins detected by mass spectrometry following co-immunoprecipitation of FLAG-tagged wild-type 52K (WT), S28/75A (A) or S28/75D (D) expressed in trans during Δ52K mutant AdV infection of HEK-293 cells. All viral proteins detected are labeled. $n = 4$ independent replicates. (G) Immunoblot validation of co-immunoprecipitation showing detection of AdV proteins (Hexon, Penton, Fiber, V, VI, IIIa, 52K, IVa2, and VII) co-precipitated with FLAG-tagged wild-type 52K (W), S28/75A (A) or S28/75D (D) expressed in trans during Δ52K mutant AdV infection of HEK-293 cells, or an IgG control. Δ52K AdV-infected HEK-293 cells without transgene expression (−) are included as a control. Immunoprecipitation is compared to input lysate. Representative of $n = 3$ independent replicates. (H–J) Co-precipitated DNA shown as a percentage of input detected after Chromatin Immunoprecipitation of FLAG-tagged wild-type 52K (WT), S28/75A (S/A), or S28/75D (S/D) expressed in trans during Δ52K AdV infection of HeLa cells. $n = 3$ independent replicates. (H) Human alpha-tubulin gene control. (I) AdV genome packaging sequence (ψ) corresponding to AT-rich repeats 1 and 2. (J) Viral DNA binding protein (DBP) open-reading frame. Data information: Data are presented as mean ± standard deviation (H–J). One-way ANOVA (H) or one-way ANOVA with Tukey's pairwise comparison tests (I, J). NS $P > 0.05$, *$P < 0.05$, **$P < 0.01$, ***$P < 0.001$, ****$P < 0.0001$. Source data are available online for this figure.

phosphorylation of 52K. We investigated the impact of the MAPK family inhibitors Selumetinib (MEK1 and MEK2) and JNK-IN-8 (JNK family), and the impact of CDK family kinase inhibitors Roscovitine (CDK2, CDK5), Dinaciclib (CDK4, CDK6), and the non-specific CDK/Crk during infection (Fig. EV2A,B). Since Dinaciclib and CDK/Crk prevented progression of infection to

the late stage, we also investigated the impact of Dinaciclib and CDK/Crk in cells in which expression of 52K is induced independent of infection (Fig. EV2C). Although downstream-target controls confirmed kinase inhibition, all inhibitors tested had no effect on the electrophoretic mobility of 52K (Fig. EV2A–C). This suggests that 52K is either phosphorylated by redundant

kinases, or by a kinase not impacted by the inhibitors used. We concluded that although the kinase or kinases responsible for the phosphorylation of 52K are as of yet unknown, it is likely phosphorylated by one or more proline-directed serine kinases.

## Phosphorylation of the N-terminal IDR does not impact viral protein interactors of the 52K protein

We next sought to establish how phosphorylation of 52K may impact its known function as a coordinator of capsid assembly and genome packaging. To accomplish this coordinating function, the 52K protein must interact, either directly or indirectly, with both viral structural proteins and with viral DNA genomes (Ahi and Mittal, 2016; Ostapchuk and Hearing, 2003, 2005). The interactions of 52K with minor capsid protein IIIa is a critical determinant of successful packaging (Ma and Hearing, 2011; Wohl and Hearing, 2008). In addition to its interaction with IIIa, the 52K protein was previously shown to co-precipitate with packaging protein IVa2, which is also essential for assembly of packaged particles (Ewing et al, 2007; Perez-Romero et al, 2005; Zhang et al, 2001; Zhang and Imperiale, 2000, 2003). To look globally at how phosphorylation of the 52K N-terminal IDR influences 52K interactions with viral proteins, we first generated 52K mutants which were phosphorylation-negative or phosphorylation-mimetic by mutating serine 28 and 75 to alanine (S28/75 A) or aspartate (S28/75D), respectively (Fig. 1E). We then performed Co-Immunoprecipitation (Co-IP) via pulldown of the FLAG-tagged 52K and examined interacting partners in an unbiased manner using Mass Spectrometry (MS). HEK-293 cells infected with Δ52K mutant AdV were complemented by ectopic expression of C-terminally FLAG-tagged WT 52K or phosphorylation mutant S28/75A or S28/75D. We then performed 52K Co-IP using anti-FLAG antibody at 24 hpi and analyzed isolated proteins by MS. Co-IP of viral capsid and packaging proteins was greater with WT or mutant 52K than with the IgG control, indicating either direct or indirect interaction with 52K (Fig. EV3A–C). In all cases, WT or mutant 52K precipitated the same virion and packaging proteins (16 proteins) at comparable levels (Figs. 1F and EV3D,E). Only the viral early protein E1B-55K, which has no known role in assembly and packaging, was enriched to a statistically significant degree compared to WT 52K, with more E1B-55K co-precipitated with the S28/75 A mutant (Fig. EV3D). The interaction of WT or mutant 52K with IIIa, IVa2, capsid protein Fiber, and core protein VI was validated by Co-IP and immunoblot (Fig. 1G). These data indicate that phosphorylation of the 52K N-terminal IDR does not influence interactions with viral structural and packaging proteins.

## Phosphorylation of the 52K IDR does not impact interaction with the viral genome

Another interaction crucial for the coordinating function of 52K is the association of 52K with AdV DNA genomes. It has been proposed that 52K protein, along with IVa2 and other packaging proteins, forms a complex associated with the packaging sequence, a cis-acting genetic element located between base pairs 200–400 at the left end of the AdV genome consisting of seven functionally redundant AT-rich repeats (Ahi and Mittal, 2016; Ewing et al, 2007; Ostapchuk et al, 2005; Perez-Romero et al, 2005, 2006). To determine whether 52K phosphorylation affects the ability of 52K to interact either directly or indirectly with viral genomes, we performed Chromatin Immunoprecipitation–quantitative Polymerase Chain Reaction (ChIP-qPCR). HeLa cells were infected with the Δ52K AdV mutant and complemented by ectopic expression of C-terminally FLAG-tagged WT 52K, phosphorylation mutants, or a GFP control. The anti-FLAG antibody was used for 52K-specific ChIP at 24 hpi and the eluted nucleic acids quantified using qPCR. We used primer pairs specific for the AdV genome packaging sequence (AT-rich repeats 1 and 2), and the AdV DNA Binding Protein (DBP) ORF, or a host genome control (alpha-tubulin). In contrast to FLAG-tagged GFP control, the WT 52K protein and both the phosphorylation-deficient and phosphorylation-mimetic mutants precipitated AdV genomes, as indicated by the detection of both the packaging sequence and the DBP ORF (Fig. 1H–J). These data confirm that phosphorylation of the 52K protein N-terminal IDR is not required for association with the AdV genome.

## Impact of phosphorylation of the 52K N-terminal IDR on condensate formation in vitro

To determine whether phosphorylation of the N-terminal IDR influences phase separation of 52K protein, we assessed condensate formation in vitro with recombinant WT or phosphorylation mutant 52K fused to an N-terminal Maltose Binding Protein (MBP) linked by a Tobacco Etch Virus (TEV) protease cleavage site. We previously showed that fusion of MBP to the N-terminal 52K IDR limits phase separation of 52K prior to the removal of the MBP tag by proteolytic cleavage (Charman et al, 2023). Recombinant proteins were expressed in *Escherichia coli*, purified using affinity chromatography, and buffer exchanged into phase-separation buffer. Incubation of the purified fusion proteins with TEV protease and analysis by SDS-PAGE and Coomassie staining confirmed the removal of the MBP tag. The released 52K proteins migrated true to their expected unmodified molecular weight of 47 kDa, consistent with the lack of 52K phosphorylation when expressed in *E. coli* (Fig. EV4A). To investigate phase separation following TEV cleavage, we assessed condensate formation by WT 52K or the S28/75A and S28/75D mutants over a range of protein concentrations (0.3125–2.5 μM) using bright-field microscopy. All three proteins formed condensates at comparable concentrations (0.625 μM and above), indicating that phosphorylation of 52K is not required to form condensates in vitro (Fig. 2A).

Phase-separated condensates may exhibit liquid-like behavior such as coalescence and surface wetting (Alberti et al, 2018, 2019; Wang et al, 2019). In contrast, maturation of condensates into a gel-like or solid-like state is associated with a loss of these liquid-like properties (Alberti et al, 2018, 2019; Wang et al, 2019). To investigate whether phosphorylation of 52K impacts the behavior of condensates formed in vitro, phase-separation assays were performed with WT 52K, S28/75A, or S28/75D proteins. The condensates were transferred to an imaging plate and observed after 20 min or 2 h, to allow condensates to settle and interact on the surface of the well. While unphosphorylated WT 52K and the S28/75A mutant settled as small discrete condensates, the phosphorylation-mimetic S28/75D mutant formed condensates that settled as larger droplets that spread out on the surface of the dish (Fig. 2B). This observation indicates a greater propensity for condensate fusion and surface wetting with the phosphorylation-mimetic S28/75D mutant compared to unphosphorylated 52K. To explore further, we again repeated in vitro

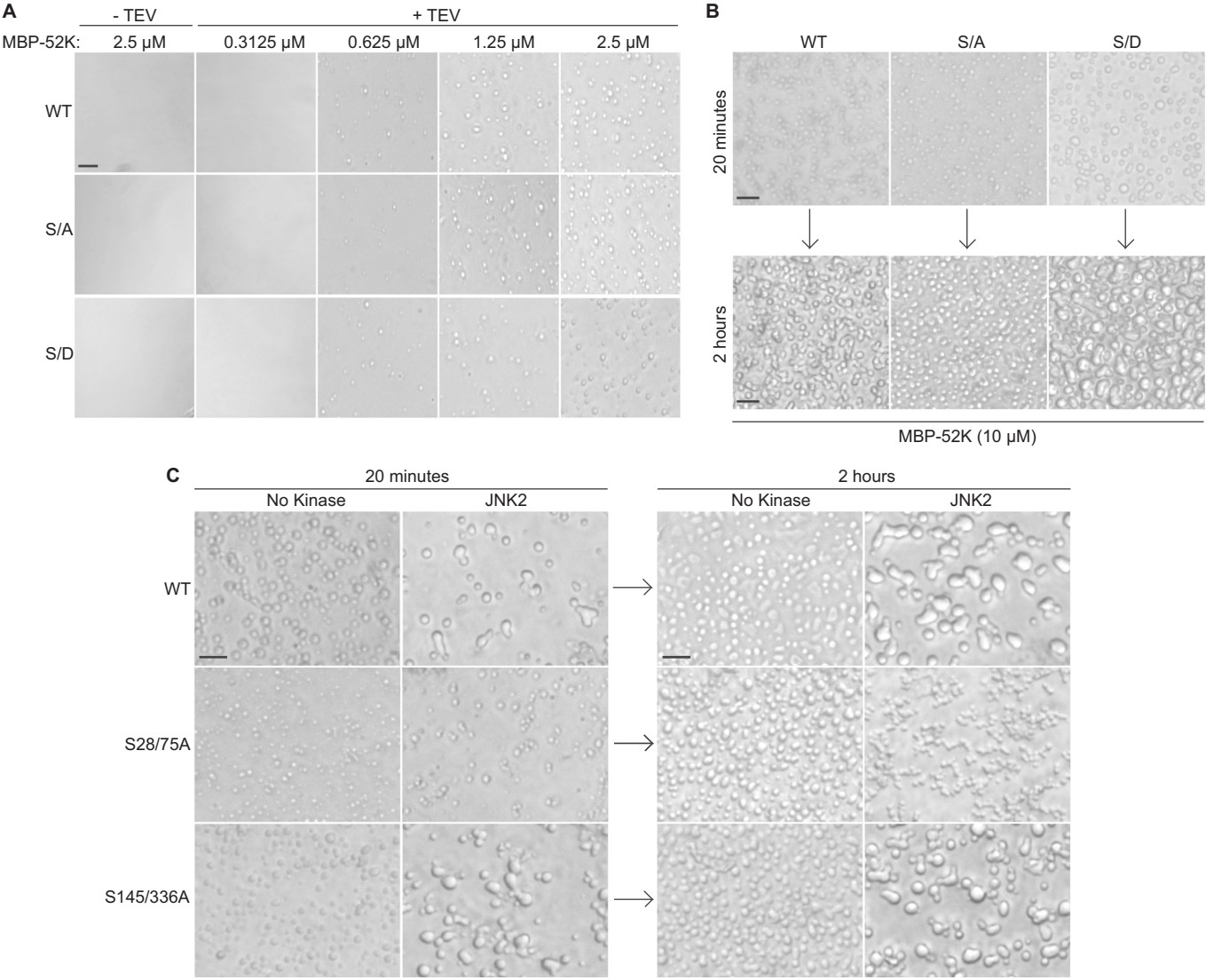

**Figure 2. Phosphorylation of the 52K IDR promotes condensate fusion and surface wetting in vitro.**

(A) Wide-field microscope images showing condensates formed by wild-type 52K (WT), or phosphorylation mutants S28/75 A (S/A) or S28/75D (S/D) in vitro at the indicated concentrations (0.3125–2.5 μM). Condensate formation was enabled by removal of the maltose binding protein (MBP) tag by addition of Tobacco Etch Virus (TEV) protease (+). A no TEV control (−) is included. Condensates were imaged after transferring to a glass-bottomed plate. Scale bars = 10 μm. Representative of n = 3 independent replicates. (B) Wide-field microscope images showing condensates formed by wild-type 52K (WT), S28/75 A (S/A), or S28/75D (S/D). Condensate formation was enabled by removal of the maltose binding protein (MBP) tag by addition of Tobacco Etch Virus (TEV) protease. Condensates were imaged after settling in a glass-bottomed plate for 20 min and then reimaged at 2 h. Each protein was assayed at a concentration of 10 μM. Scale bars = 10 μm. Representative of n = 3 independent replicates. (C) Wide-field microscope images showing condensates formed by wild-type 52K (WT), S28/75 A, or S145/336A. MBP-52K fusion proteins were incubated with or without JNK2 kinase for 1 h before the addition of TEV protease. Condensates were imaged after settling in a glass-bottomed plate for 20 min and then reimaged at 2 h. Each protein was assayed at a concentration of 10 μM. Scale bars = 10 μm. Representative of n = 3 independent replicates. Source data are available online for this figure.

phase-separation assays following in vitro phosphorylation of 52K by the CMGC group MAPK JNK2 (MAPK9). To investigate whether phosphorylation of the N-terminal IDR rather than phosphorylation of the C-terminus is responsible for this phenotype, we compared WT 52K and S28/75A to an additional mutant in which serine residues 145 and 336, which were identified as phosphorylated in our MS analysis, were mutated to alanine (S145/336A). Treatment of WT 52K or the S145/336A mutant with

JNK2 resulted in reduced electrophoretic mobility indicative of phosphorylation, while treatment of S28/75A did not (Fig. EV4B), indicating that serine residues 28 and 75 are phosphorylated by JNK2 in vitro. Phosphorylation of WT 52K increased coalescence and surface wetting compared to the unphosphorylated control, while treatment of the S28/75A mutant did not (Fig. 2C), indicating that these liquid-like characteristics are dependent on phosphorylation of serine residues in the IDR. Phosphorylation of the S145/

336A mutant phenocopied phosphorylation of WT 52K (Fig. 2C), indicating that these changes in condensate behavior do not require phosphorylation of residues S145 and S336. We conclude that phosphorylation of the 52K protein N-terminal IDR promotes the liquid-like characteristics of condensates formed in vitro.

To investigate whether the observed differences in condensate properties may result from altered oligomerization, we assessed the oligomerization status of 52K by running a native (non-denaturing) PAGE of our recombinant MBP-tagged proteins with or without removal of the MBP tag by TEV protease cleavage. Without removal of the MBP tag, all proteins were resolved as both higher molecular weight oligomers greater than 1048 kDa and lower molecular weight oligomers consistent with the expected size of MBP-52K dimers (~180 kDa). Removal of the MBP tag resulted in a greater retention of 52K in the well, likely resulting from condensate formation. The 52K protein formed higher molecular weight oligomers that exhibited heterogeneous progression through the gel, as well as lower molecular weight oligomers consistent with the expected size of 52K dimers (~94 kDa) (Fig. EV4C). These data suggest that 52K forms a dimer capable of higher-order oligomerization, and that this oligomerization is not dependent on its phosphorylation status. We concluded that observed differences in the condensate behavior when comparing unphosphorylated 52K compared with phosphomimic S28/75D mutant do not result from altered dimerization. We propose that differences in condensate behavior instead result from subtler changes in the dynamic interactions mediated by the N-terminal IDR.

## Phosphorylation of the 52K N-terminal IDR influences nuclear body morphology in cells

To investigate whether phosphorylation of the N-terminal IDR influences the formation of 52K BMCs inside cells, we ectopically expressed WT 52K or phosphorylation mutants in stably transduced A549 cells in which transgene expression is induced by addition of doxycycline. All proteins were expressed to comparable levels following induction, as evidenced by immunoblotting. Altered electrophoretic mobility suggested that mutation of S28 and S75 to alanine successfully prevented phosphorylation of 52K (Fig. 3A). Visualization of 52K by immunofluorescence–confocal microscopy revealed that expression of all proteins resulted in NB formation (Fig. 3B), with comparable numbers of cells exhibiting NBs in each case (Fig. 3C). Both WT 52K and the S28/75D mutant formed NBs that were typically small and numerous, while the S28/75A mutant formed fewer, larger NBs (Fig. 3D,E). In all cases, we observed a high degree of cell-to-cell variation in NB number and size (Figs. 3D,E and EV5A), likely due to variation in expression levels as previously observed (Charman et al, 2023). Transient expression of WT 52K or phosphorylation mutants in HEK-293 cells phenocopied our observations in transgenic A549 cells (Fig. EV5B–F). We conclude that phosphorylation of 52K alters the morphology of NBs, promoting smaller, more numerous NBs.

## Phosphorylation of the 52K N-terminal IDR increases sensitivity of nuclear bodies to 1,6-hexanediol

Given that PTMs of IDRs can regulate IDR-IDR interactions (Bah and Forman-Kay, 2016; Darling and Uversky, 2018; Iakoucheva

et al, 2004; Owen and Shewmaker, 2019), we reasoned that phosphorylation of the 52K protein IDR may regulate IDR-IDR interactions, which in turn would impact BMC properties. The aliphatic alcohol 1,6-hexanediol is thought to interfere with weak multivalent interactions, and is able to disrupt many BMCs when present at sufficient concentrations (Alberti et al, 2018, 2019; Kroschwald et al, 2017). We therefore asked whether 1,6-hexanediol is sufficient to disrupt NBs assembled from WT or phosphorylation mutant 52K. We induced 52K expression in transgenic A549 cells and waited 10 h to allow NBs to form before incubating cells either with or without 400 mM (4.72% w/v) 1,6-hexanediol for 10 min prior to fixation. We then imaged the cells by immunofluorescence–confocal microscopy and analyzed the percentage of cells under each condition that formed 52K NBs. Addition of 1,6-hexanediol did not affect overall protein levels (Fig. 3F). However, addition of 1,6-hexanediol did effectively disrupt NBs formed by WT 52K or the phosphorylation-mimetic S28/75D mutant, with only 19.7% ± 0.4 SD (WT) or 27.2% ± 4.1 SD (S28/75D) of cells exhibiting NBs compared to untreated controls where the percentage of cells with nuclear bodies was high (>92%) for all cell lines (Fig. 3G,H). In contrast, NBs formed by the S28/75A mutant were more resistant to disruption by 1,6-hexanediol, with 86.1% ± 0.8 SD of 1,6-hexanediol-treated cells exhibiting NBs (Fig. 3G,H).

We next assessed whether a low concentration of 1,6-hexanediol is sufficient to prevent the formation of NBs. We induced expression of 52K and exchanged the culture media at 8 h post induction and incubated the cells for a further 16 h with either 75 mM (0.885% w/v) 1,6-hexanediol or control media without 1,6-hexanediol. Under these conditions, the cells were alive, and cell membranes were intact (Fig. EV6A). Although levels of WT and mutant 52K protein at 24 h post induction were slightly reduced in 1,6-hexanediol-treated cells relative to control cells, they remained comparable between all treated cell lines as evidenced by immunoblot (Fig. 3I). At 8 h post induction, the levels of 52K were low, as were the percentage of cells with NBs (WT = 6.2% ± 2.3 SD, S28/75A = 21.8% ± 4.2 SD, S28/75D = 4.5% ± 1.5 SD) (Fig. 3J,K). By 24 h post induction, the percentage of cells with NBs was >90% in control cells not treated with 1,6-hexanediol, indicating that NBs formed during this incubation period (Fig. 3J,K). In contrast, very few cells formed NBs when cells expressing WT 52K (5.6% ± 2.3 SD) or the S28/75D mutant (13.0% ± 0.3 SD) were incubated with 1,6-hexanediol (Fig. 3J,K), indicating that low concentrations of 1,6-hexanediol is sufficient to attenuate NB formation in these cell lines. However, 80.2% ± 2.4 SD of cells expressing the S28/75A mutant formed NBs (Fig. 3J,K), indicating that NB formation of the phosphorylation-deficient S28/75A mutant is more resistant to antagonism by 1,6-hexanediol than WT 52K and the phosphorylation-mimetic S28/75D mutant. Equivalent experiments with propylene glycol mirrored results observed with 1,6-hexanediol, although higher concentrations of propylene glycol were required to antagonize NBs than with 1,6-hexanediol (Fig. EV6B–H). The addition of 1,6-hexanediol to in vitro assays, also demonstrated the ability of this diol to antagonize condensate formation. Higher concentrations of 1,6-hexanediol were required to prevent condensate formation of the WT 52K and the S28/75A mutant compared to the S28/75D mutant (Fig. EV4D,E), indicating that unphosphorylated forms of 52K are more resistant

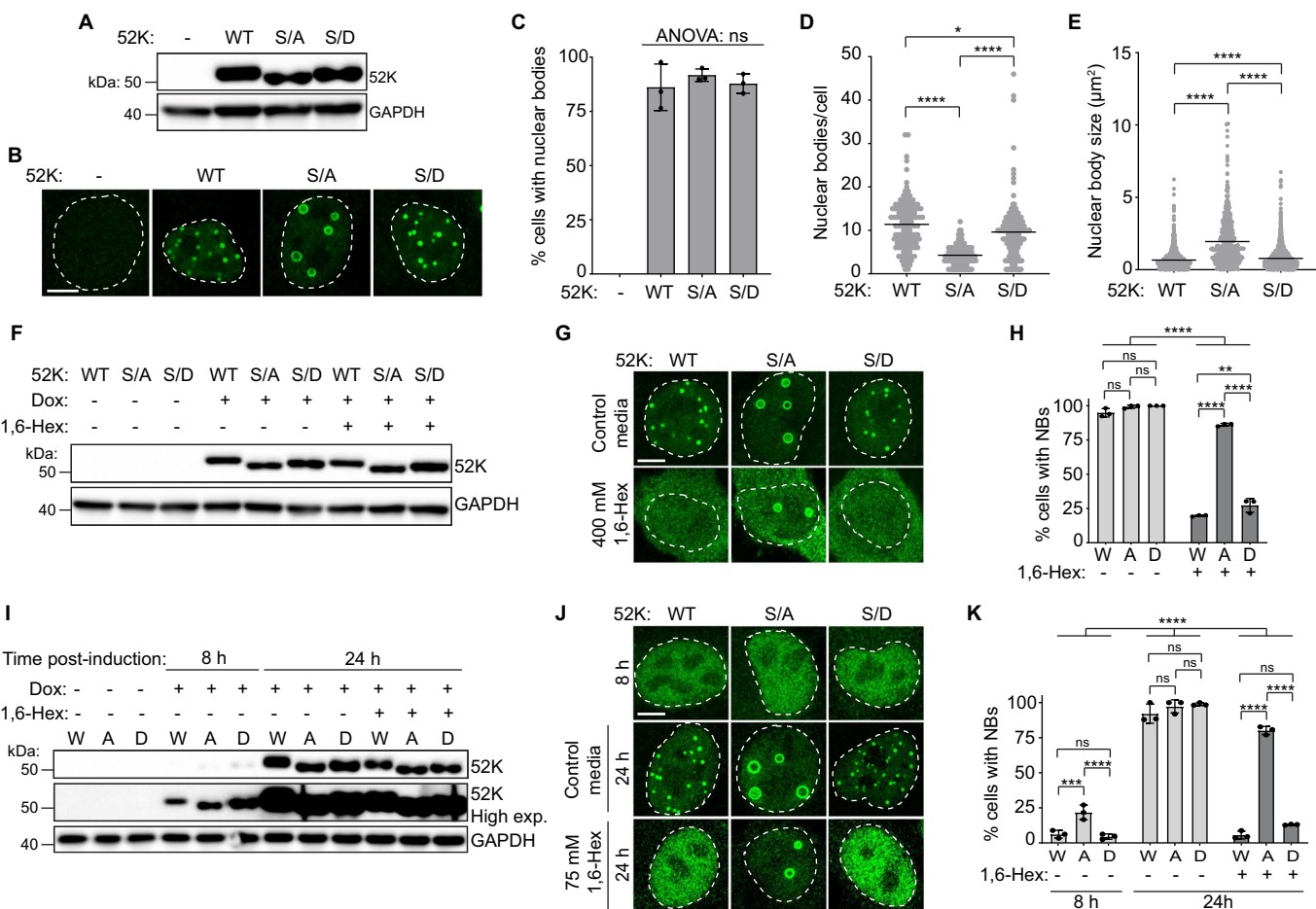

**Figure 3. Phosphorylation of the 52K IDR impacts nuclear body morphology and sensitivity to 1,6-hexanediol.**

(A–E) Parental (−) or transgenic A549 cell lines expressing wild-type 52K (WT), or phosphorylation mutants S28/75A (S/A), or S28/75D (S/D) at 24 h after induction of expression by addition of doxycycline. (**A**) Immunoblot showing levels of 52K. GAPDH is shown as a loading control. Representative of n = 3 independent replicates. (**B**) Immunofluorescence–confocal microscope images showing the morphology of 52K nuclear bodies. Nuclei outlined (dashed white line). Scale bar = 5 μm. Representative of n = 3 independent replicates. (**C**) Percentage of cells with nuclear bodies. n = 3 independent replicates each consisting of three fields of view analyzed. (**D**) Number of nuclear bodies per nucleus. n = 3 independent replicates pooled. A total of 117 (WT), 108 (S/A), or 61 (S/D) nuclei were analyzed. (**E**) Nuclear body size. n = 3 independent replicates pooled. A total of 1330 (WT), 459 (S/A), or 1548 (S/D) nuclear bodies were analyzed. (**F–H**) Expression and nuclear bodies in transgenic A549 cells. Expression of wild-type 52K (WT), S28/75A (S/A), or S28/75D (S/D) was induced by addition of doxycycline for 10 h before the addition of 400 mM 1,6-hexanediol for 10 min prior to analysis. (**F**) Immunoblot showing levels of 52K. GAPDH is shown as a loading control. Representative of n = 3 independent replicates. (**G**) Immunofluorescence–confocal microscope images showing the localization of 52K. Nuclei are outlined (dashed white line). Scale bar = 5 μm. Representative of n = 3 independent replicates. (**H**) Percentage of cells with nuclear bodies. n = 3 independent replicates each consisting of three fields of view analyzed. (**I–K**) Transgenic A549 cells. Expression of wild-type 52K (WT), S28/75 A (S/A), or S28/75D (S/D) was induced by addition of doxycycline for 8 h before incubation with control media or media containing 75 mM 1,6-hexanediol for an additional 16 h (24 h post induction). (**I**) Immunoblot showing levels of 52K. High exposure panel shows the lower levels of expression at 8 h. GAPDH is shown as a loading control. Representative of n = 3 independent replicates. (**J**) Immunofluorescence–confocal microscope images showing the localization of 52K. Nuclei are outlined (dashed white line). Scale bar = 5 μm. Representative of n = 3 independent replicates. (**K**) Percentage of cells with nuclear bodies. n = 3 independent replicates each consisting of 3 fields of view. Data information: Data are presented as mean ± standard deviation (**C, H, K**) or mean and individual data points (**D, E**). One-way ANOVA (**C**), Kruskal–Wallis ANOVA with Dunn's pairwise comparison tests (**D, E**), or two-way ANOVA with Tukey's pairwise comparison tests (**H, K**). NS P > 0.05, *P < 0.05, **P < 0.01, ***P < 0.001, ****P < 0.0001. Source data are available online for this figure.

to antagonism than the phosphomimic. Together these experiments suggest that the disruption of 52K nuclear bodies by 1,6-hexanediol and propylene glycol is likely due to direct effects of these diols on multivalent interactions rather than off-target effects. We conclude that NBs formed by 52K are less sensitive to antagonism by 1,6-hexanediol in the absence of phosphorylation at S28 and S75, consistent with phosphorylation of the 52K protein IDR acting as a regulator of the dynamic multivalent interactions that underpin these nuclear BMCs.

## Phosphorylation of the 52K N-terminal IDR promotes dynamic exchange and internal rearrangement of 52K

BMCs held together by weak, transient, multivalent interactions typically exhibit dynamic exchange of resident proteins between BMCs and the wider soluble phase (Alberti et al, 2019; Shin and Brangwynne, 2017). Accordingly, we reasoned that regulation of IDR-IDR interactions by 52K phosphorylation may in turn influence this dynamic exchange. To investigate this hypothesis,

we expressed GFP-tagged WT or phosphorylation mutant 52K in HEK-293 cells and performed Fluorescent Recovery After Photobleaching (FRAP) (Fig. 4A). To assess dynamic exchange with the wider nucleoplasm, we photobleached punctate NBs ($<1\ \mu m^2$) and measured their recovery over a period of 22.5 s. NBs formed by WT

52K recovered fluorescence rapidly ($T_{1/2} = 2.2\ s \pm 0.8\ SD$) (Fig. 4B–D; Movie EV1). This high degree of dynamic exchange was also observed for NBs formed by the S28/75D mutant ($T_{1/2} = 2.7\ s \pm 1.2\ SD$) (Fig. 4B–D; Movie EV2). However, NBs formed in cells expressing the S28/75A mutant recovered slower

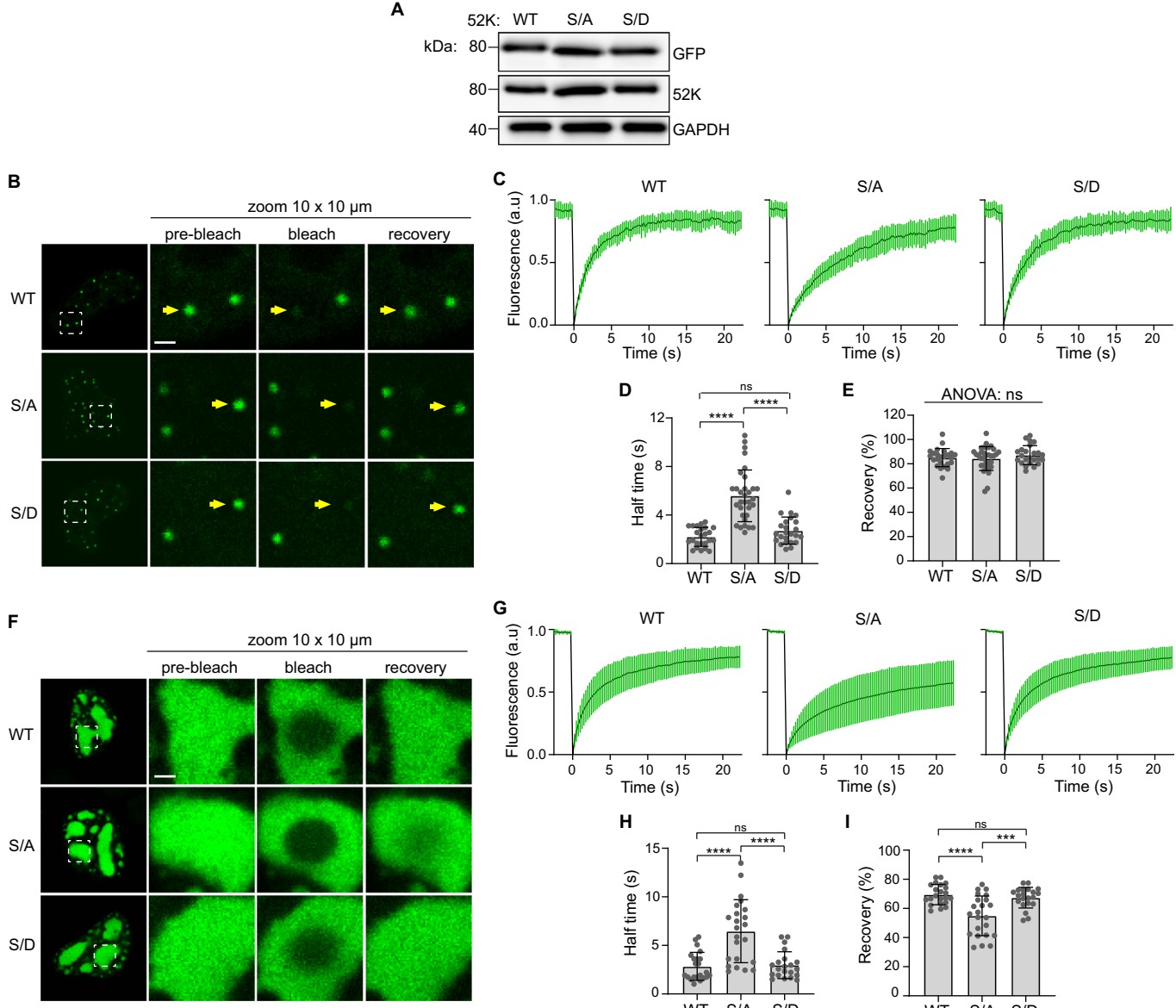

**Figure 4. Phosphorylation of the 52K IDR promotes dynamic exchange and internal mobility of 52K.**

(A) Immunoblot showing ectopic expression of 52K in HEK-293 cells expressing wild-type 52K (WT), S28/75A (S/A), or S28/75D (S/D) tagged with GFP at their C-terminus. GAPDH is shown as a loading control. Representative of $n = 3$ independent replicates. (B–E) Fluorescence recovery after photobleaching of punctate nuclear bodies in HEK-293 cells expressing GFP-tagged wild-type 52K (WT) or phosphorylation mutants S28/75A (S/A), or S28/75D (S/D). $n = 3$ independent replicates pooled. A total of 24 (WT), 32 (S/A), or 24 (S/D) nuclear bodies were analyzed. (B) Representative live cell confocal images showing punctate nuclear bodies. The dashed white line indicates $10 \times 10\ \mu m$ zoomed area. Zoomed area panels show nuclear bodies 1 s before photobleaching (pre-bleach), immediately after photobleaching (bleach), or 10 s after photobleaching (recovery). Targeted nuclear bodies are indicated by yellow arrows. Scale bar $= 2\ \mu m$. (C) Fluorescence recovery curves. (D) Recovery half times. (E) Percentage fluorescence recovery. (F–I) Fluorescence recovery after internal photobleaching (4.155 $\mu m$ bleach spot diameter) of large nuclear bodies in HEK-293 cells expressing GFP-tagged wild-type 52K (WT), S28/75A (S/A), or S28/75D (S/D). $n = 3$ independent replicates pooled. A total of 22 (WT), 23 (S/A), or 21 (S/D) nuclear bodies were analyzed. (F) Representative live cell confocal images showing large nuclear bodies. The dashed white line indicates $10 \times 10\ \mu m$ zoomed area. Zoomed area panels show nuclear bodies 1 s before photobleaching (pre-bleach), immediately after photobleaching (bleach), or 20 s after photobleaching (recovery). Scale bar $= 2\ \mu m$. (G) Fluorescence recovery curves. (H) Recovery half times. (I) Percentage fluorescence recovery. Data information: Data are presented as mean ± standard deviation (B–D, F–H). One-way ANOVA with Tukey's pairwise comparison tests (C, D, G, H). NS $P > 0.05$, ***$P < 0.001$, ****$P < 0.0001$. Source data are available online for this figure.

($T_{1/2}$ = 5.6 s ± 2.1 SD) (Fig. 4B–D; Movie EV3). Total recovery was comparable in all cases (Fig. 4E). We conclude that phosphorylation of S28 and S75 promotes rapid exchange of 52K between NBs and the nucleoplasm.

We anticipated that a loss of dynamic IDR-IDR interactions in favor of more stable, fixed interactions should manifest as a reduction in internal rearrangement of 52K within BMCs. To investigate the internal rearrangement of 52K within BMCs, we capitalized on the fact that larger, atypical NBs form in a subset of high-expressing cells following transient expression of 52K-GFP (Charman et al, 2023), making it possible to bleach target regions within these NBs. We found that 52K is mobile within larger NBs, with WT 52K ($T_{1/2}$ = 2.8 s ± 1.4 SD, recovery = 69% ± 7 SD) and the S28/75D mutant ($T_{1/2}$ = 3.0 s ± 1.4 SD, recovery = 67% ± 7 SD) both exhibiting relatively rapid and high total recovery (Fig. 4F–I; Movies EV4 and 5). However, slower rates of recovery and lower total recovery were observed in cells expressing the S28/75 A mutant ($T_{1/2}$ = 6.5 s ± 3.2 SD, average recovery = 55% ± 14 SD) (Fig. 4F–I; Movie EV6). We conclude that the mobility of 52K within NBs is restricted in the absence of phosphorylation.

## Phosphorylation of the 52K N-terminal IDR promotes AdV infectious progeny production

We next sought to determine how phosphorylation of the 52K protein IDR impacts the assembly of infectious progeny particles. Although 52K is essential for assembly of packaged particles, it is not required for genome replication (Charman et al, 2023; Gustin and Imperiale, 1998). A comparison of WT AdV and the Δ52K mutant AdV genome replication by qPCR confirmed equivalent genome replication (Fig. 5A). Since 52K is dispensable for genome replication, infectious progeny production of a the Δ52K AdV mutant can be specifically complemented by expression of 52K in trans (Charman et al, 2023; Gustin and Imperiale, 1998). We therefore investigated whether progeny production of the Δ52K mutant virus could be complemented by transient expression of the S28/75A and S28/75D mutants in HEK-293 cells. To provide comparative context, we also infected HEK-293 cells with WT AdV. Immunoblot analysis revealed that the abundance of WT and mutant 52K proteins during complementation were similar, allowing for each condition to be compared (Fig. 5B). Infection with WT AdV in the absence of exogenous 52K produced ~$10^9$ focus forming units (FFU) of infectious virus, while Δ52K AdV produced no infectious progeny as anticipated (Fig. 5C). Expression of WT 52K or the S28/75D mutant restored infectious progeny production of the Δ52K virus to the order of $10^7$ FFU, whereas expression of the S28/75A mutant failed to restore infectious progeny production to levels that could be distinguished from background resulting from input virus (~$10^5$ FFU) (Fig. 5C).

To expand on these findings, we employed an orthogonal approach to complement Δ52K infection, using our transgenic A549 cells in which expression of WT 52K or phosphorylation mutants is induced by the addition of doxycycline. Expression levels were comparable in all 52K rescue assays (Fig. 5D). Expression of WT 52K or the S28/75D mutant restored infectious progeny production of the Δ52K virus to the order of $10^8$ FFU (Fig. 5E). This level of progeny production was higher than that achieved when 52K was expressed by transient transfection, likely due to the more homogenous levels of 52K achieved in inducible transgenic cells. In this

system, expression of the S28/75A mutant restored Δ52K AdV infectious progeny production to the order of $10^7$ FFU (Fig. 5E). This suggests that although the S28/75A mutant can contribute to infectious progeny production to some low level, this ability is attenuated compared to phosphorylated or phosphorylation-mimetic 52K. Analysis of viral genome abundance by qPCR demonstrated comparable genome replication in Δ52K AdV cells expressing WT 52K, S28/75A, or S28/75D (Fig. 5F), indicating that differences in infectious progeny production does not result from differences in genome replication. We conclude that phosphorylation of the 52K protein N-terminal IDR promotes AdV progeny production.

## Phosphorylation of the 52K N-terminal IDR promotes AdV genome packaging

We next sought to determine whether the absence of S28 and S75 phosphorylation results in fewer complete packaged particles being generated. We complemented Δ52K infection by expressing WT 52K or our phosphorylation mutants in trans using our inducible system, and then harvested and purified resulting viral particles using Cesium Chloride (CsCl) density gradient ultracentrifugation. The number of particles purified was assessed using the spectrophotometry method (Maizel et al, 1968). Cells infected with the Δ52K AdV without complementation produced only incomplete particles, which are less dense (1.29 g/cc) than packaged particles (1.34 g/cc) and are therefore resolved as a higher band on CsCl gradients (Alba et al, 2007; Charman et al, 2023; Condezo and San Martín, 2017) (Fig. 5G). In contrast, packaged particles that resolve as a lower band on CsCl gradients were purified from cells expressing 52K, although fewer packaged particle were assembled in the presence of the phosphorylation-deficient S28/75A mutant ($4.5 \times 10^3$) compared to WT 52K ($1.6 \times 10^4$) or the S28/75D mutant ($1.7 \times 10^4$) (Fig. 5G,I). All infections produced comparable numbers of incomplete particles per infected cell ($1.8 \times 10^3$) (Fig. 5H).

Analysis of purified particles by SDS-PAGE and immunoblotting confirmed that the packaged particles contained processed forms of genome-associated core protein VII and core protein VI (Fig. 5J), which are known to be proteolytically cleaved by AdV protease during packaging (Kulanayake and Tikoo, 2021; Mangel and Martín, 2014; Pérez-Berná et al, 2014). In contrast, incomplete particles were associated with low levels of unprocessed core proteins, and a greater abundance of packaging proteins (Fig. 5J). As expected, packaging proteins L4-100K and 52K, which are not components of the mature particle, were abundant in only the incomplete particles (Fig. 5J). Interestingly, incomplete particles purified from Δ52K-infected cells complemented with the S28/75A mutant were associated with more 52K than incomplete particles from Δ52K-infected cells expressing WT 52K or the S28/75D mutant (Fig. 5J). This may indicate attempts to package viral genomes which generate failed assembly intermediates enriched for 52K. We also note that the defect in the assembly of packaged particles observed during complementation with S28/75A was smaller than the defect in infectious progeny production we observed using the same system of transgene expression. This suggests the possibility of additional assembly defects that cannot be distinguished by CsCl gradient ultracentrifugation or SDS-PAGE analysis of particle composition. Taking these findings together, we conclude that phosphorylation of S28 and S75 in the 52K protein IDR promotes the assembly of infectious, packaged viral particles.

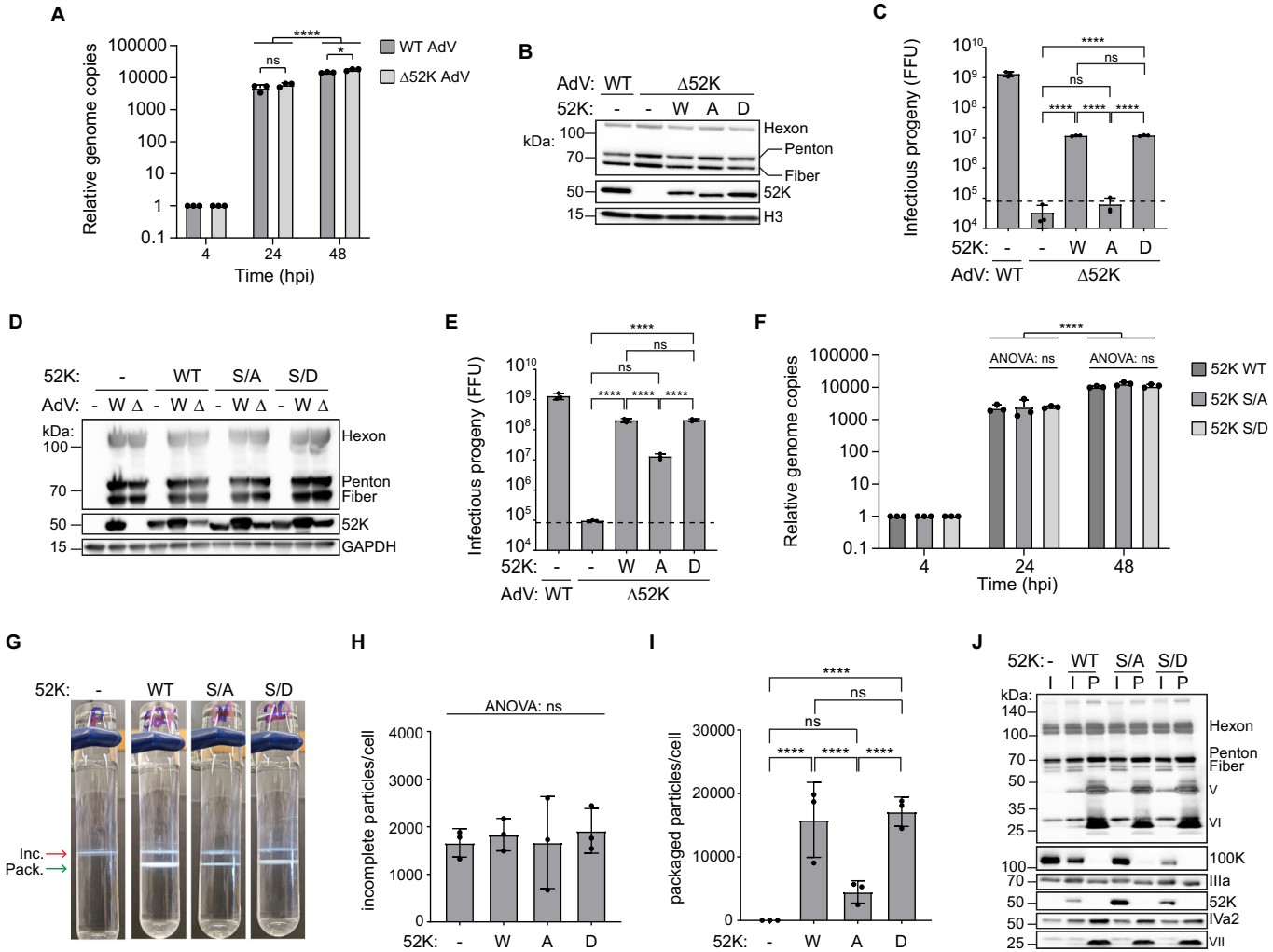

**Figure 5. Phosphorylation of the 52K IDR promotes AdV infectious progeny production.**

(A) Comparison of viral genome replication in A549 cells infected with wild type (WT) or Δ52K mutant adenovirus (AdV). Samples were harvested at 4-, 24-, or 48-h post infection (hpi). 4 hpi represents input prior to genome replication. Relative genome copy number was determined by quantitative polymerase chain reaction. $n = 3$ independent replicates. (B, C) HEK-293 cells without transfection (−) or transiently expressing wild-type 52K (W) or phosphorylation mutants S28/75A (A) or S28/75D (D) were infected with wild-type (WT) or Δ52K mutant adenovirus (AdV). $n = 3$ independent replicates. (B) Representative immunoblot showing levels of AdV-late proteins (Hexon, Penton, Fiber) and 52K. Histone H3 is shown as a loading control. (C) Infectious progeny production measured as focus forming units (FFU). (D–J) Parental (−) or transgenic A549 cells were infected with WT or Δ52K mutant AdV or mock infected as a control (−) as indicated. Expression of wild-type 52K (WT or W), S28/75A (S/A or A) or S28/75D (S/D or D) was induced by addition of doxycycline at 2 h post infection. of $n = 3$ independent replicates. (D) Representative immunoblot showing levels of AdV-late proteins (Hexon, Penton, Fiber) and 52K. GAPDH is shown as a loading control. (E) Infectious progeny production measured as focus forming units (FFU). (F) Comparison of Δ52K mutant AdV genome replication Samples were harvested at 4-, 24-, or 48 hpi. The 4 hpi time point represents input prior to genome replication. Relative genome copy number was determined by quantitative polymerase chain reaction. (G) Representative images showing Cesium Chloride (CsCl) density gradient purified bands of incomplete (Inc.) and packaged (Pack.) progeny particles. (H) Quantification of incomplete viral particles per infected cell isolated by CsCl density gradient purification. (I) Quantification of packaged viral particles per infected cell isolated by CsCl density gradient purification. (J) Representative immunoblot showing levels of AdV capsid (Hexon, Penton, Fiber, IIIa), Core (V, VI, VII), and Packaging (52K, IVa2, 100K) proteins in purified incomplete (I) and packaged (P) CsCl bands. $1 \times 10^9$ particles were analyzed for each condition. Data information: Data are presented as mean ± standard deviation (A, C, E, F, H, I). Two-way ANOVA with Tukey's multiple comparison's tests (A, F). One-way ANOVA (H) or one-way ANOVA with Tukey's pairwise comparison tests (C, E, I). NS $P > 0.05$, *$P < 0.05$, ****$P < 0.0001$. Source data are available online for this figure.

## Discussion

The biophysical properties and behavior of BMCs depend on the nature of the dynamic multivalent interactions that underpin their formation. The loss of dynamic interactions favors more stable interactions linked with the transition to a more gel-, glass-, or solid-like state (Brangwynne et al, 2015; Flock et al, 2014; Harmon

et al, 2017; Holehouse and Pappu, 2018; Lin et al, 2015; Oldfield and Dunker, 2014; Patel et al, 2015; Riback et al, 2017; Uversky, 2019). This has raised questions as to how the network of interactions that enable BMC formation are regulated in cells to ensure that condensates maintain the properties required for their biological function. Using AdV infection as a model system, we show that phosphorylation of the viral 52K protein modulates

condensate properties and promotes the assembly of infectious viral progeny.

Recent advances demonstrate that a number of phase-separating viral proteins are modified by phosphorylation (Carlson et al, 2020; Etibor et al, 2021; Geiger et al, 2021; Guseva et al, 2020; Hirai et al, 2021; Papa et al, 2019; Su et al, 2021). Phosphorylation of phase-separating proteins has largely been studied in the context of molecular switches that promote the assembly or disassembly of BMCs, or the recruitment of key factors (Aumiller and Keating, 2016; Hofweber and Dormann, 2019; Kwon et al, 2013; Owen and Shewmaker, 2019; Snead and Gladfelter, 2019; Yang et al, 2020). In this study, we demonstrate that phosphorylation of the AdV 52K protein at residues S28 and S75 modulates viral BMCs, promoting dynamic liquid-like condensate properties. These serine residues fall within the N-terminal IDR of 52K, previously demonstrated to be critical for BMC formation and viral progeny production (Charman et al, 2023). Specifically, S28 sits within a proline-rich spacer region proposed to act as a fluidizing entity, whereas S75 sits between charged motifs that facilitate IDR-IDR interactions required for phase separation. By comparing BMC formation in cells expressing either WT 52K, or phosphorylation-negative (S28/75 A) or mimetic (S28/75D) mutants, we found that phosphorylation of the 52K protein IDR is not required for BMC formation. However, BMCs formed in the absence of phosphorylation at S28 and S75 exhibited modified properties compared to phosphorylated WT 52K and the phosphorylation-mimetic mutant. This included decreased sensitivity to 1,6-hexanediol and propylene glycol, decreased exchange of 52K with the surrounding nucleoplasm, and decreased internal rearrangement of 52K within larger BMCs. Our findings in vitro further support a role for phosphorylation as a regulator of BMC properties, with condensates formed by both phosphorylation-mimetic 52K and WT 52K phosphorylated by JNK2 kinase exhibiting more fluid-like properties compared to unmodified WT 52K or the phosphorylation-negative S28/75A mutant. These findings are consistent with the loss of dynamic IDR-IDR interactions that underpin 52K BMC formation, in favor of more stable interactions associated with the transition to a more gel-, glass-, or solid-like state (Brangwynne et al, 2015; Choi et al, 2020; Holehouse and Pappu, 2018; Lin et al, 2015).

We recently showed that the AdV 52K protein is essential for the formation of BMCs that organize viral capsid proteins (Charman et al, 2023). We demonstrated that the organization of viral capsid proteins into BMCs allows for capsid assembly that is coordinated with the provision of viral genomes required to assemble packaged progeny (Charman et al, 2023). It follows that localization of 52K and capsid proteins to BMCs occurs upstream of their incorporation into particles, and subsequent particle maturation which releases 52K from viral particles (Hasson et al, 1989, 1992; Pérez-Berná et al, 2014). This model of BMC-mediated assembly predicts that for progeny production to remain viable, viral capsid proteins organized at BMCs must be maintained in a state that is conducive to downstream assembly. When we investigated the ability of 52K expressed in trans to rescue progeny production of a mutant virus that does not express 52K, we found that the phosphorylation-deficient S28/75A mutant was attenuated in its ability to support coordinated progeny production when compared to WT 52K or the phosphorylation-mimetic S28/75D, and this resulted in the assembly of fewer packaged particles. We propose that phosphorylation of the 52K protein N-terminal IDR

plays an important role in regulating the IDR-IDR interactions that underpin viral BMCs. In this way, phosphorylation would maintain the network of dynamic multivalent interactions that allow for 52K and viral capsid protein to be concentrated within these membrane-less sub-compartments while remaining viable for assembly that is synchronized with the provision of viral genomes. Conversely, stabilization of IDR-IDR interactions that result in phenotypic condensate hardening may compromise viral assembly by immobilizing 52K and recruited capsid proteins within BMCs. However, we do not rule out the possibility that phosphorylation could also play a more direct role in maintaining specific dynamic IDR-IDR interactions that mediate the assembly of packaged viral particles from these organized assembly components.

It should be noted that a number of viral proteins that play roles in packaging and assembly are known to be phosphorylated, including reported phase-separating proteins (Carlson et al, 2020; Etibor et al, 2021; Guseva et al, 2020; Lu et al, 2021; Risso-Ballester et al, 2021). For example, the rotavirus non-structural protein 5 (NSP5) is critical for the formation of viral replication factories, cytoplasmic compartments that harbor viral assembly (Geiger et al, 2021). NSP5 is hyperphosphorylated late in infection, corresponding with the recruitment of viral structural proteins, the production of viral progeny, and the loss of liquid-like properties associated with viral factories (Papa et al, 2019). It was also recently shown that phosphorylation of the SARS-CoV2 nucleocapsid protein promoted liquid-like condensate properties in vitro (Carlson et al, 2020). Given that nucleocapsid protein plays a role in both genome replication and packaging, it was proposed that phosphorylation may promote genome replication, while the unphosphorylated form may enable packaging (Carlson et al, 2020). These findings suggest that different viral processes may benefit from different condensate properties, and that timely switching of phosphorylation status may be critical for viral progeny production. Consistent with the notion that functional condensates require specific properties, modulating the properties of viral factories during Respiratory Syncytial Virus inhibited progeny production (Risso-Ballester et al, 2021).

In contrast to many RNA viruses which organize both genome replication and viral assembly within the same viral factories, AdV BMCs that organize 52K and capsid proteins do not harbor viral genome replication, suggesting that they instead play a dedicated role in assembly and packaging (Charman et al, 2023; Charman and Weitzman, 2020; Hidalgo and Gonzalez, 2019). This makes it possible to assess directly how condensate properties impact these processes during AdV infection. Our finding that AdV progeny production requires liquid-like condensate properties shares similarities with experiments performed during infection with Influenza A virus (IAV). IAV ribonucleoprotein complexes form condensates that are thought to play a role in the assembly of progeny particles (Alenquer et al, 2019). It was recently reported that drug-induced oligomerization of the viral nucleoprotein, which is thought to stabilize interactions between ribonucleoproteins, diminishes the liquid-like properties of viral condensates. This modulation of condensate properties referred to as "hardening" correlated with compromised progeny production (Etibor et al, 2023). This suggests that at least in the case of certain viruses, successful assembly of packaged progeny particles specifically requires assembly components to be organized and maintained in a liquid-like condensate.

In conclusion, our data demonstrate that phosphorylation of the adenovirus L1-52/55 kilodalton protein regulates viral BMCs to promote assembly of packaged progeny particles in AdV-infected cells. These findings indicate that host-cell phosphorylation machinery is redirected for the benefit of viral assembly and highlight the importance of IDR posttranslational modification as a means of maintaining the dynamic condensate properties required for condensate functionality. This contribution furthers our understanding of the complexity of viral assembly and provides a framework to better understand how the biophysical properties of BMCs may enable or compromise their role in viral processes. In addition, our findings highlight the exciting possibility of targeting host-cell pathways as a means to attenuate viral progeny production via modulation of viral BMC properties.

# Methods

## Cell culture

All cell lines were obtained from and authenticated by the American Type Culture Collection (ATCC). ATCC authentication of cell lines includes assessment of cell morphology, karyotyping, and short tandem repeat profiling. Cells were cultured under standard conditions (37 °C and 5% $CO_2$) in sterile, TC-treated, nonpyrogenic, polystyrene tissue culture dishes (Cat#: Corning 430167). HEK-293 (Cat#: ATCC CRL-1573), HEK-293T (Cat#: ATCC CRL-3216), and HeLa (Cat#: ATCC CCL-2) cells were grown in DMEM (Corning, Cat#: 10-013-CV) supplemented with 10% FBS (VWR, Cat#: 89510-186) and 1% Pen/Strep (Gibco, Cat#: 15140-122). A549 cells (Cat#: ATCC CCL-185) were maintained in Ham's F-12 Kaighn's Modification medium (Gibco, Cat#: 21127-022) supplemented with 10% FBS and 1% Pen/Strep. All cell lines tested negative for mycoplasma using the LookOut Mycoplasma PCR Detection Kit (Sigma-Aldrich). Mycoplasma testing was carried out on 1 mL of cell culture media taken from tissue culture dishes containing confluent monolayers of cells on a routine basis at least twice a year.

## Transgenic cell lines

The stable transgenic cell lines for doxycycline-inducible expression of WT 52K, S28/75A, or S28/75D were generated by lentivirus transduction and puromycin selection of A549 lung adenocarcinoma cells. Lentivirus was generated by co-transfection of HEK-293T cells with lentivirus expression plasmid and two helper plasmids, pMD.G2 and pCMVΔR8.74 (see "Plasmids and transfections"), and the cells cultured in media containing serum that was heat inactivated at 56 °C for 30 min. Cell supernatant was collected at 48 and 72 h and filtered through 0.45-μM Acrodisc syringe filters (VWR, Cat#: 28143-312) using a 10-mL syringe. Cells were transduced by incubation with lentivirus supernatant in the presence of 10 μg/mL polybrene (Santa Cruz, Cat#: sc-134220) for 8 h. The supernatant was replaced with fresh lentivirus supernatant for an additional 16-h transduction. Lentivirus supernatant was then replaced with fresh culture media. Cells were cultured for a further 24 h before initial selection in 5 μg/mL puromycin (Gibco, cat#: A1113802). Cells were allowed to grow to confluency once under 5 μg/mL puromycin, before being shifted to a maintaining concentration of 1 μg/mL for all experiments.

## Plasmids

To generate mammalian expression plasmids, open-reading frames were PCR amplified and inserted into parent plasmids by restriction digest and ligation. All restriction enzymes were purchased from New England Biolabs and were compatible with digest reactions in rCutsmart buffer (Cat#: New England Biolabs B6004S). All plasmid ligations were done with T4 DNA ligase purchased from New England Biolabs (Cat#: New England Biolabs M0202S). The untagged WT 52K open-reading frames were used in a pEYFP-C1 (Clontech Cat#: 6006-1) background in place of the EYFP open-reading frame (Charman et al, 2023). Phosphorylation mutant 52K open-reading frames were generated through site-directed mutagenesis of the WT 52K open-reading frame in the pEYFP-C1 background using primer pairs specific to each mutant. Primer Sequences were as follows:

S28A F: GCACCCGCCCCTCCTCCTACCGCGTCAGGAG.
S28A R: AGGAGGGGCGGGTGCCCTGCATGTCTGCCG.
S75A F: GCGCCCGCTCCTGAGCGGTACCCAAGGGTGC.
S75A R: CTCAGGAGCGGGCGCTCCTAGCCGCGCCAG.
S145A F: GGGATTGCTCCCGCGCGCGCACACGTGGCG.
S145A R: CGCGGGAGCAATCCCGGTTCGCGCGTCGGG.
S336A F: GAGCTCGCCGACCGCGAGCTGATGCACAGCCTG.
S336A R: GCGGTCGGCGAGCTCGCGCCGCCGGCTCAC.
S28D F: GCACCCGACCCTCCTCCTACCGCGTCAGGAG
S28D R: AGGAGGGTCGGGTGCCCTGCATGTCTGCCG
S75D F: GCGCCCGATCCTGAGCGGTACCCAAGGGTGC
S75D R: CTCAGGATCGGGCGCTCCTAGCCGCGCCAG.

For expression of MBP-tagged fusion proteins in bacteria, 52K open-reading frames were PCR amplified from the pEYFP-C1 background using primers and inserted into pMAL-c2X (Addgene, plasmid#: 75286) using a fragment assembly strategy and Gibson Assembly Master Mix (NEB, Cat#: E2611S) following the manufacturer's instructions (Charman et al, 2023). Primer sequences were as follows:

52K forward: ATTCGGATCCTCTAGAGAAAACCTGTACTT CCAGGGACATCCGGTGCTGCGGCAG
52K reverse: GCAGGTCGACTCTAGATTAGTGGTGATGGT GATGATGGTACTCGCCGTCCTCTGGC.

For fusion proteins tagged with GFP, open-reading frames were excised from the pEYFP-C1 background and inserted into pEGFP-N1 (Clontech Cat#: 6085-1) containing a L221K mutation to prevent dimerization of GFP molecules using a XhoI/XbaI double-restriction enzyme approach (New England Biolabs Cat#s: R0146S/R0145S). pLKO.dCMV.TetO/R was a gift from Chris Boutel (Busnadiego et al, 2014). Lentivirus expression plasmids encoding untagged forms of WT 52K or 52K phosphorylation mutants were generated by excising open-reading frames from the pEYFP-C1 background using a NheI/SalI double-restriction enzyme approach (New England Biolabs Cat#s: R3131S/R0138S). Open-reading frames were then inserted into the multiple cloning site of pLKO.dCMV.TetO/R. Lentivirus plasmids pMD.G2 (plasmid #12259) and pCMVΔR8.74 (plasmid #22036) were purchased from Addgene. C-terminal monomeric FLAG-tagged forms of WT 52K or 52K phosphorylation mutants were generated by excising open-reading frames from the pEYFP-C1 background using a XhoI/BamHI double-restriction enzyme approach (New England Biolabs Cat#s: R0146S/R0136S). Open-reading frames were then inserted into the multiple cloning site of c-FLAG pcDNA3 (Cat#: Addgene

20011). Plasmids generated as part of this study were checked by a combination of diagnostic restriction digest followed by DNA agarose gel visualization, Sanger sequencing completed by Genewiz (Azenta Life Sciences), and whole-plasmid nanopore sequencing completed by Plasmidsaurus to confirm correct insertion of the open-reading frames.

## Antibodies

The following primary antibodies for viral proteins were used: Anti-adenovirus Type 5 antibody raised against whole adenovirus capsids; recognizing late proteins Hexon, Penton, Fiber, and proteins V, VI, VII, VIII, and IX (Cat#: Abcam ab6982), species: rabbit, polyclonal, WB 1:10,000. Antibody to 52K (gift from P. Hearing) (Ostapchuk et al, 2005), species: rabbit, polyclonal, WB 1:10,000, IF 1:300. Antibody to IIIa (gift from P. Hearing) (Ma and Hearing, 2011), species: rabbit, polyclonal, WB 1:5000, IF 1:300. Antibody to L4-100K (gift from P. Hearing) (Yan et al, 2016), species: rabbit, polyclonal, WB 1:5000. Antibody to Protein VII (gift from H. Wodrich) (Johnson et al, 2004; Komatsu et al, 2015), species: rabbit, polyclonal, WB 1:2000, IF 1:300. Antibody to IVa2 (gift from P. Hearing) (Ostapchuk et al, 2005), species: rabbit, polyclonal, WB 1:2000, IF 1:300.

The following primary antibodies for cellular proteins were used: GFP (Abcam, Cat#: ab290, Lot: GR3321614-1), species: rabbit, polyclonal, WB 1:5000. GAPDH (GeneTex, Cat#: GTX100118, Lot: 43712), species: rabbit, polyclonal, WB 1:5000. Histone H3 (EMD Millipore, Cat#: 06-755, Lot: 24721), species: rabbit, polyclonal, WB 1:5000. FLAG (Sigma-Aldrich, Cat#: F3165, Lot: SLCJ3741) species: mouse, clone M2, Immunoprecipitation 5 μg/IP, chromatin- immunoprecipitation 5 μg/ChIP. FLAG (Sigma-Aldrich, Cat#: 7425, Lot: 078M4886V) species: rabbit, polyclonal, WB 1:5000. pRSK1 T359/S363 (Abcam, Cat#: ab32413, Lot: GR149751-14) species: rabbit, clone: E238, WB 1:1000. pERK1/ 2 T185/Y187 (Thermo-Fisher, Cat#: 44-680, Lot: 2434425) species: rabbit, polyclonal, WB 1:1000. pC-Jun S63 (Abcam, Cat#: ab32385, Lot: GR3339196-2) species: rabbit, clone: Y172, WB 1:1000. pJNK1/ 2 T183/Y185 (Thermo-Fisher, Cat#: 44-682, Lot: 2465206) species: rabbit, polyclonal, WB 1:1000. pATF2 S71 (Abcam, Cat#: ab32019, Lot: GR3343221-3) species: rabbit, clone: E268, WB 1:1000. pMAPKAPK2 T334 (Cell Signaling Technology, Cat#: 3007T, Lot: 12). species: rabbit, clone: 3007, WB 1:1000. pP38 T181/Y183 (Abcam, Cat#: ab195049, Lot: 1000887-3) species: rabbit, clone: 18120, WB 1:1000. Cyclin D1 (Abcam, Cat#: ab134175, Lot: GR3393633-14) species: rabbit, clone: EPR2241, WB 1:1000. pRNA Polymerase II CTD Repeat YSPTSPS S5 (Abcam, Cat#: ab5131, Lot: GR3462844-1) species: rabbit, polyclonal, WB 1:1000.

For immunoblot analysis, Horseradish peroxidase-conjugated (HRP) goat anti-mouse (Jackson Laboratories, Cat#: 115-035-003) or goat anti-rabbit (Jackson Laboratories, Cat#: 111-035-045) secondary antibodies were used at a concentration of 1:5000 and 1:10,000, respectively. For immunofluorescence, Alexa Fluor fluorophore-conjugated secondaries were used at a concentration of 1:1000. The following fluorophores were used: goat anti-rabbit 488 (Life Technologies, Cat#: A-11008), goat anti-mouse 488 (Life Technologies, Cat#: A-11001), goat anti-mouse 555 (Life Technologies, Cat#: A-21422), goat anti-rat 555 (Life Technologies, Cat#: A-21434) and goat anti-rabbit 647 (Life Technologies, Cat#: A-21245).

## Viruses and infections

Human adenovirus type C5 (Ad5) wild-type (WT) was purchased from ATCC (Cat#: VR-5), propagated on HEK-293 cells, purified via cesium chloride density ultracentrifugation, and stored in 40% glycerol at −20 °C for infections. The Ad5 Δ52K mutant pm8001 (Δ52K) (Gustin and Imperiale, 1998), a gift from P. Hearing (Stony Brook University, NY), was propagated on transgenic cell line (A549) expressing WT 52K, purified via cesium chloride density ultracentrifugation, and stored in 40% glycerol at −20 °C for infections. Viruses were purified using two sequential rounds of ultracentrifugation in cesium chloride density gradients. To achieve a cryoprotective solution for storage, the virus was diluted as follows: 2-parts virus in cesium chloride, 1-part 5× viral dilution solution (40 mM Tris pH 8, 400 mM NaCl, 0.4% BSA in $H_2O$), 2-parts 100% glycerol. Viral titers were determined by infectious focus forming assay as described in "Viral progeny production". All infections were carried out using a multiplicity of infection (MOI) of 10 unless stated otherwise and harvested at indicated hours post infection (hpi). To infect cells, the virus was diluted in cell culture media without FBS. After 2 h at 37 °C, culture media containing 10% FBS was added. For virus yield assays, the virus infection media was removed after 2 h and cells were washed 1× in PBS before the addition of culture medium to remove the excess virus.

## Kinase inhibitors

The following kinase inhibitors were purchased from Selleck Chemicals as lyophilized powders, resuspended at 10 mM stock concentrations in DMSO, and stored at −20 °C. Selumetinib (AZD 6244 or ARRY-142886, Cat#: S1008, Lot: S100835), JNK-IN-8 (JNK Inhibitor VIII, Cat# S4901, Lot: S490101), Dinaciclib (SCH727965, Cat#: S2768, Lot: S276807). The following kinase inhibitors were purchased from Millipore Sigma as lyophilized powders, resuspended at 10 mM stock concentrations in DMSO, and stored at −20 °C. Roscovitine (Cat#: 186692-46-6, Lot: 3876029). CDK/CRK inhibitor (Cat#: 784211-09-2, Lot: 3285036). For use in cells, the virus inoculum or induction inoculum was removed and replaced with fresh media supplemented with 10 μM of inhibitor or matched DMSO (vehicle) control. For experiments using doxycycline-inducible transgenic A549 cells, media containing inhibitor or DMSO also contained fresh doxycycline. Unless indicated otherwise, inhibitors were added at 2 h post infection/induction. Infection/induction was allowed to progress in the presence of inhibitors or DMSO for 24 h before being harvested for immunoblot.

## Cell lysate phosphatase treatment

A549 and HEK-293 cells were cultured in 12-well nonpyrogenic, polystyrene tissue culture dishes, infected with wild-type Ad5 at a multiplicity of 10, and harvested at 24 h post infection. Cells were harvested in 400 μL of homemade lysis buffer identical in composition to the New England Biolabs NEBuffer for Protein MetalloPhosphatases (50 mM HEPES pH 7.5, 1% Triton X-100, 100 mM NaCl, 1 mM $MnCl_2$, in $H_2O$). Lambda Protein Phosphatase (1 μL or 100 units) (New England Biolabs, Cat#: P0753S) was added to 44 μL of protein lysate and incubated for 30 min at 30 °C. In total, 4× LDS sample buffer (15 μL) was added to quench the

reaction and prepare lysate for loading onto SDS-PAGE. Protein lysate (20 μL) was loaded onto SDS-PAGE and subjected to the immunoblot protocol for protein detection.

## SDS-PAGE, Coomassie staining, and immunoblot analysis

SDS-PAGE and immunoblot analysis were carried out using standard methods. In brief, protein samples were prepared using lithium dodecyl sulfate (LDS) loading buffer (Thermo-Fisher, Cat#: NP0007) supplemented with 25 mM Dithiothreitol (DTT) and boiled at 95 °C for 10 min. Equal amounts of protein lysate were separated by SDS-PAGE. For Coomassie, gels were stained with 0.1% w/v brilliant blue (Sigma-Aldrich, Cat#: B0149) in 10% v/v acetic acid and 40% v/v methanol for 10 min before de-staining in 10% v/v acetic acid and 40% v/v methanol. For immunoblotting, proteins were transferred onto methanol activated Polyvinylidene Fluoride (PVDF) membrane (Millipore Sigma, Cat#: IPFL00010) at 30 V for 60–120 min and blocked in blocking buffer (5% milk in TBST supplemented with 0.05% sodium azide) for 1 h at room temperature. Membranes were incubated overnight at 4 °C with primary antibodies diluted in blocking buffer, washed for 30 min in TBST, incubated for 1 h at room temperature with HRP-conjugated secondary antibody diluted in blocking buffer, and washed again for 30 min in TBST. Proteins were visualized with Pierce ECL Western Blotting Substrate (Thermo-Fisher, Cat#: 34577) and detected using a Syngene G-Box using GeneSys acquisition software. Images were processed and assembled in Adobe Illustrator CS6.

## Bioinformatics and predicted structure

The Ad5 52K protein (L1 packaging protein 3) amino acid sequence was obtained from Uniprot (Q6VGV2). Disorder tendency was analyzed using the IUPred3 algorithm (https://iupred3.elte.hu/) using the default settings (Erdos et al, 2021). Phospho-peptide sequences for the 52K S28 and S75 phospho-sites were provided to Kinexus Bioinformatics Corporation (Kinexus). Phospho-peptide sequences were 15 amino acids long and centered on the S28 and S75 phosphorylated residues. Phospho-peptide sequences were as follows: 52K S28: RQTCRAPSPPPTASG, 52K S75: LARLGAPSPERHPRV. Kinexus, using their published algorithm(Safaei et al, 2011), conducted an In Silico Phosphoprotein Match Prediction (IPMP) screen to identify kinases in the human kinome that were likely to recognize the S28 and S75 sites of 52K as substrates. 52K structures were generated by inputting the primary amino acid sequence of the 52K protein into Alpha-fold 2.0 (Jumper et al, 2021; Ruff and Pappu, 2021). The resulting .PDB files generated by AlphaFold 2.0 were downloaded and then input into the University of California San Francisco's Chimera Extensible Molecular Modeling System to generate modifiable structures (Pettersen et al, 2004).

## Transgene expression in mammalian cells

For transient expression of transgenes, mammalian expression plasmids were transfected into HEK-293T or HEK-293 cells using X-tremeGENE HP (Roche, Cat#: 6366236001), following the manufacturer's instructions for a 3:1 reagent:plasmid ratio.

Plasmids were transfected at a ratio of 1 μg plasmid: 3 μL X-tremeGene HP: 4e5 cells and scaled up or down accordingly for all experiments. For expression and localization independent of infection, cells were analyzed 24 h post-transfection. For co-transfection, both plasmids were transfected at equal concentrations. For complementation of Δ52K virus infection by transient transfection, cells were infected and then subsequently transfected 2 hpi. For stable cell lines, transgene expression was induced by addition of doxycycline at a final concentration of (1 μg/ml). For transgene expression during infection, expression was induced 2 hpi.

## Mass spectrometry (MS)

In preparation for mass spectrometry analysis, 52K Co-IP samples were neutralized with 8 M urea in 50 mM Ammonium Bicarbonate pH 8, reduced with 5 mM dithiothreitol (DTT) for 1 h at room temperature, alkylated using 10 mM iodoacetamide (IAM) for 45 min in the dark and 15 min in the light, followed by addition of trypsin at a 1:50 ratio overnight at room temperature. Samples were then reconstituted and desalted using homemade C18 stage tips. Samples were run on a standard linear 2-h gradient using standard proteomics buffers of 0.1% formic acid in $H_2O$ and 0.1% formic acid in 80% acetonitrile (ACN). Samples were quantified using a Thermo QE-HF™ MS instrument and batch-randomized to account for instrument variation. The DDA MS method was designed with the MS1 having a window of 330–1100 $m/z$, AGC target of 1e6 and MIT of 75 ms with the MS2 having automated windows, AGC target of 100% and MIT of 75 ms. The selection for ions were charges 2–8, minimum peak intensity of 1e4, and a 3 s maximum cycle time.

Thermo raw files were processed using Proteome Discoverer 2.4 using Sequest to identify peptide spectral matches (PSMs). Files were first processed in Sequest using Uniprot human and an in house Ad5 FASTA for protein identifications. Sequest settings were 10 ppm for precursor and 0.02 Da for fragment thresholds for peak assignment. The FDR for both was set to 1% for PSM identifications. The PTMs were filtered for site specific localization of 90% or higher with the ptmRS tool (Taus et al, 2011). Peptides that passed these filters were then filtered again with a 1% FDR for peptide and protein assignments. Data was then further processed in R for presentation. For all sample comparisons, two-sided Student's $T$ tests were used to determine significant changes with a $P$ value cutoff of 0.05 and a fold enrichment cutoff of 2. Co-IP MS was repeated independently four times to ensure reproducibility. Only hits detected in all four independent replicates were considered valid.

## Co-immunoprecipitation (Co-IP)

For Co-IP, HEK-293 cells were infected at MOI 10 with Δ52K Ad5 for 2 h at 37 °C, 5% $CO_2$ in DMEM without FBS or Penn/Strep. At 2 hpi, cells were washed once in PBS to remove excess virus and topped with fresh DMEM + 10% FBS. Cells were then transfected for 6 h with 20 μg (10-cm tissue culture dish) of mammalian expression plasmid encoding C-terminally FLAG-tagged WT or phosphorylation mutant 52K proteins. After 6 h, transfection inoculum was removed and replaced with fresh DMEM + 10% FBS + 1% Penn/Strep. At 24 hpi, cells were harvested, washed in

PBS, pelleted, and freshly lysed in 400 µL of Lysis Buffer (20 mM Tris pH 8, 137 mM NaCl, 1% Igepal, 2 mM EDTA, 1× HALT protease and phosphatase inhibitor cocktail (Thermo-Fisher, Cat#: 78440)) in Eppendorf Protein Lo-Bind tubes (Eppendorf, Cat#: 0030108434). Cell lysates were lysed on ice for 1 h, followed by sonication at 4 °C using a Diagenode Biorupter on the low frequency for six cycles of 30 s on/30 s off. Samples were clarified by centrifugation at 12,000× g for 10 min at 4 °C. Protein concentrations for each lysate were determined by comparison to a standard curve of BSA protein via BCA assay using the Pierce BCA protein assay kit (Thermo-Fisher, Cat#: 23225). For IP of WT or phosphorylation mutant 52K, 5 µg of mouse anti-FLAG or mouse IgG isotype control was added to 500 µg of protein from clarified cell lysate and rotated overnight at 4 °C. For bead purification, 35 µL of Protein G Dynabeads (Thermo-Fisher, Cat#: 10004D) per IP were washed in IP Lysis Buffer and added to IPs, followed by rotation at 4 °C for 2 h. Beads were then washed 4× in IP Lysis Buffer followed by 4× washes with IP Wash Buffer (10 mM Tris pH 7.4, 1 mM EDTA, 150 mM NaCl, 1× HALT). For immunoblot analysis, samples were eluted in 50 µL of 1× LDS sample buffer supplemented with 25 mM fresh DTT, boiled for 10 min at 95 °C, and analyzed by SDS-PAGE. İn total, 1% of input lysates were compared to 10% of Co-IP elutions (IgG and FLAG). For mass spectrometry elution, preparation, and analysis, see "Mass spectrometry". IPs were repeated in independent quadruplicate.

## Chromatin immunoprecipitation (ChIP)

For ChIP, HeLa cells were infected at MOI 10 with Δ52K Ad5 for 2 h at 37 °C, 5% CO$_2$ in DMEM without FBS or Penn/Strep. At 2 hpi, cells were washed once in PBS to remove excess virus and topped with fresh DMEM + 10% FBS. Cells were then transfected for 6 h with 20 µg (10-cm tissue culture dish) of C-terminally FLAG-tagged WT or phosphorylation mutant 52K proteins. After 6 h, transfection inoculum was removed and replaced with fresh DMEM + 10% FBS + 1% Penn/Strep. At 24 hpi, cells were washed once in PBS and resuspended in fresh DMEM. Cells were then cross-linked using 1% formaldehyde final concentration for 10 min at room temperature. Cross-linking reactions were quenched with 1 M glycine on ice for 5 min, followed by washing the cells 1× in PBS. Cells were pelleted and subjected to cytoplasmic removal using nuclei isolation buffer (10 mM Tris pH 7.5, 10 mM NaCl, 0.1% Igepal, 1× HALT) on ice for 10 min. Intact nuclei were isolated by centrifugation at 3000× g for 5 min at 4 °C. Intact nuclei were subjected to lysis in 0.3% SDS at 37 °C for 1 h, followed by the addition of 2% tritonX-100 at 37 °C for 1 h. Host chromatin and viral genomes liberated from lysed nuclei were subjected to enzymatic digestion using the KpnI – HF and SmaI restriction enzymes from New England Biolabs in 1× rCutsmart overnight at 37 °C. The remaining nuclear debris was pelleted away by centrifugation at 12,000× g for 10 min at 4 °C, and clarified supernatants were used for subsequent IPs. For IPs and bead purification, 35 µL of Protein G Dynabeads were washed 3× in PBST (0.02% Tween 20) and pre-bound to mouse anti-FLAG or mouse IgG isotype control antibodies by rotating at room temperature for 1 h. Chromatin from clarified nuclear lysate (20%) was resuspended in 965 µL of ChIP Dilution Buffer (0.01% SDS, 1.1% tritonX-100, 1.2 mM EDTA, 16.7 mM Tris pH 8, 167 mM NaCl, 1× HALT) and 35 µL of pre-bound Protein G

Dynabeads were added. IPs were rotated at 4 °C for 4 h. Beads were then washed 1× in Low Salt Wash (0.1% SDS, 1% TritonX-100, 2 mM EDTA, 20 mM Tris pH 8, 150 mM NaCl), 1× in High Salt Wash (0.1% SDS, 1% TritonX-100, 2 mM EDTA, 20 mM Tris pH 8, 500 mM NaCl), 1× in Lithium Chloride Wash (1% Igepal, 1% Sodium Deoxycholate, 1 mM EDTA, 10 mM Tris pH 8, 250 mM LiCl), and 2× in TE buffer. DNA was eluted from beads in 100 µL of elution buffer (1% SDS, 10 mM EDTA, 50 mM Tris pH 8) by incubating at 65 °C, 900 rpm for 30 min in a thermomixer. Input chromatin (10%) was also resuspended in 100 µL of elution buffer, and formaldehyde cross-links were reversed overnight at 65 °C, 900 rpm in a thermomixer. Contaminating RNA was degraded from elution by incubation with RNAse A (80 µg) (Qiagen, Cat#: 19101) at 37 °C, 900 rpm for 2 h in a thermomixer. Contaminating protein was degraded from elution by incubation with Proteinase K (80 µg) (Thermo-Fisher, Cat#: 25-530-049) at 55 °C, 900 rpm in a thermomixer. The remaining DNA was purified from elutions into 100 µL of nuclease-free water using Agencourt AMPure XP Beads (Beckman-Coulter, Cat#: A63880) following the manufacturer recommended ratio of 1.8× Beads: Elution Volume. Elutions were further diluted 1:10 in nuclease-free water and stored at −20 °C for qPCR.

## Quantitative PCR (qPCR)

For analysis of DNA co-precipitated with 52K, qPCR was performed on DNA eluted from ChIP experiments. Cycle threshold values for each experimental condition were normalized to cycle threshold values for the corresponding 10% input condition and presented as percent enrichment relative to input. For increased accuracy, ChIP-qPCRs were performed in technical triplicate for three independent replicates.

For analysis of viral genome replication, qPCR was performed on DNA harvested from AdV5 or Δ52K AdV-infected A549 cells. Infected cells were harvested by scraping, pelleted by centrifugation, and frozen in liquid nitrogen. DNA was extracted from frozen cell pellets using the PureLINK™ Genomic DNA Mini Kit (Invitrogen, Cat# K182002, Lot# 2535435) by following the standard protocol with one modification. Samples were heated at 56 °C for 30 min instead of 10 min before purification to disrupt AdV particles within cells and release AdV genomes for quantification. Copies of the AdV DBP amplicon were quantitated using the Livak ΔΔCT method with DBP as the target gene and alpha-tubulin as the housekeeping gene. Data are presented as relative genome copies (DBP copies) on a log10 scale.

All qPCR reactions were performed in 384-well plates using 10 µL volumes, with reaction components as follows: 3 µL of diluted purified DNAs (1:10 dilution), 5 µL SYBR Green PCR Master Mix(Applied Biosystems, Cat#: 4309155), 0.1 µL of forward primer, 0.1 µL of reverse primer, 1.8 µL of nuclease-free, biological grade water. qPCR Primers were specific for regions within the Ad5 DBP and cellular alpha-tubulin ORFs as previously described (Pancholi and Weitzman, 2018; Price et al, 2020). In addition, primers for the Ad5 genomic packaging sequence AT Rich Repeats A1/A2 were used. Primer Sequences were as follows for these primers: forward A1/A2 Primer: TGACAATTTTCGCGCGGTTTT, reverse A1/A2 Primer: CGCGCTATGAGTAACACAAAT. All qPCR reactions were performed on the Applied Biosystems Viia 7 Real-Time PCR System operating on QuantStudio 7 software using SYBR Green

PCR Master Mix and following the manufacturer's instructions for standard speed reactions. For increased accuracy, each independent replicate was performed in technical triplicate.

## Purification of recombinant proteins

For expression of MBP fusion proteins, Bl21 De3 LysS cells (New England Biolabs, Cat#: C2527H) were transformed with bacterial expression plasmids by heat shock at 42 °C and selected on LB agar plates containing chloramphenicol, carbenicillin, and 0.5% w/v glucose for 16 h at 37 °C. Colonies were used to inoculate LB broth containing chloramphenicol, carbenicillin, and 0.5% glucose and cultured at 37 °C in a shaking incubator until an optical density (600 nm) of 0.4–0.6 was reached. Expression was induced by the addition of 0.5 mM IPTG (Roche, Cat#: 10724815001) for 16 h at 15 °C. Cultures were pelleted by centrifugation and frozen prior to purification. Pellets were thawed/resuspended in lysis/binding buffer supplemented with Halt Protease Inhibitor cocktail (Thermo-Fisher, Cat#: 78445). The lysates were incubated with Benzonase (Sigma-Aldrich, Cat#: E1014) at a final concentration of 250 Units/mL for 1 h at 4 °C before sonication (×6 cycles of 20 s on —20 s off on "low" setting in 4 °C water bath), and the lysates clarified by centrifugation. Protein was purified by affinity purification using Express Ni resin (New England Biolabs, Cat#: S1428S) following the manufacturer's guidelines with the following modifications. Express Ni resin was pre-equilibrated with Ni lysis/binding buffer (20 mM sodium phosphate, 500 mM NaCl, 1 mM DTT, pH 8), washed in Ni wash buffer (20 mM sodium phosphate, 300 mM NaCl, 2.5% glycerol, 5 mM Imidazole, 1 mM DTT, pH 8), and eluted in Ni elution buffer (20 mM sodium phosphate, 300 mM NaCl, 2.5% glycerol, 500 mM Imidazole, 1 mM DTT, pH 8). Purification was performed at 4 °C to limit protein degradation. Purity of protein was confirmed to be >90% as assessed by SDS-PAGE and Coomassie staining. All purified proteins or labeled purified proteins were sub-aliquoted and stored at −80 °C to limit protein degradation. Protein concentration was determined by Bradford assay using Pierce Bradford reagent (Thermo-Fisher, Cat#: 23246) and NanoDrop 2000c spectrophotometer (Thermo-Fisher).

## Buffer exchange and concentration of purified proteins

Purified proteins were buffer exchanged and concentrated using Amicon 10 kDa spin filters (EMD Millipore, Cat#: UFC801008) using the manufacturer's guidelines. For buffer exchange, protein samples were concentrated a minimum of 20-fold prior to re-dilution in the exchange buffer. Concentration and re-dilution were repeated a total of three times to achieve a dilution factor in excess of 8000.

## In vitro condensate formation assays

In vitro condensate formation assays were carried out in phase-separation buffer (25 mM Tris-HCl, 150 mM NaCl, 1 mM DTT, pH 7.4) using the indicated concentration of proteins in a final volume of 35 μL. TEV protease (10 units) (New England Biolabs, Cat#: P8112S) was added, and the reaction was incubated at room temperature for 1 h prior to transfer of 30 μL to a 384-well glass bottom plate. Condensates were allowed to settle for 5 min prior to

imaging using an EVOS imaging system (Thermo Fischer Scientific). Unless indicated otherwise, assays were imaged within a 15-min window. Protein samples were centrifuged to remove any debris, and concentration was checked prior to use.

## In vitro phosphorylation of recombinant 52K

In vitro kinase assays were performed on purified, full-length MBP-52K fusion proteins using His-JNK2 (His-MAPK9) (Thermo, Cat# PV3620). Proteins (10 μM) were provided as substrates for 20 μg/mL of His-JNK2, and reactions were performed for 1 h at room temperature in 35 μL total volume in PCR tubes. Reactions were performed in standard phase-separation buffer (25 mM Tris-HCl pH 7.4, 150 mM NaCl) supplemented with 10 mM MgCl$_2$ and 10 mM ATP. Reactions were terminated with 1× NuPAGE LDS Sample Buffer supplemented with 25 mM DTT and boiled at 95 °C for 10 min to prepare for SDS-PAGE. Phosphorylation of 52K proteins was observed by molecular weight up-shift after SDS-PAGE Coomassie staining.

## Native-PAGE

Non-denaturing native conformation polyacrylamide gel electrophoresis was carried out on purified MBP-52K fusion proteins using the NativePAGE™ Sample Preparation Kit (Invitrogen, Cat# BN2008) and NativePAGE™ 3–12% Bis-Tris polyacrylamide gels (Invitrogen, Cat# BN1003BOX) in the presence of 1× NativePAGE™ Coomassie G-250 Cathode Buffer Additive (Invitrogen, Cat# BN2002) as per the manufacturer's instructions. Purified protein was prepared in 1× Native-PAGE sample buffer (Invitrogen, Cat# BN2003) supplemented with 0.2% Coomassie G-250 sample additive and separated until the dye front was at the bottom of the gel. Proteins were then visualized by immunoblot methods as previously described.

## In vitro kinase assays (Kinexus)

In vitro kinase assays were carried out by Kinexus Bioinformatics Corporation (Kinexus) according to the standard reaction conditions for the company using the ADP-Glo™ Kinase Assay platform developed and provided by Promega Corporation (Promega, Cat#: V6930). Synthetic peptides (500 μM) were provided as substrates for the 25 human kinases listed, and reactions were performed for 30 min at 30 °C in 25 μL total volume in 96-well plates according to the manufacturer specified protocol (Promega, Cat# V6930). Reactions were terminated with ADP-Glo™ reagent (Promega, Cat# V6930), and relative light units were measured with a GloMax™ plate reader (Promega, Cat# E7031) after incubation with Kinase Detection Reagent™ (Promega, Cat# V6930) for 30 min at room temperature(Zegzouti et al, 2010). Substrate peptide sequences were as follows: 52K S28: CRAPSPPPT, 52K S75: LGAPSPERH.

## Immunofluorescence–confocal microscopy

Cells were grown on 12-mm glass coverslips (Electron Microscopy Sciences, Cat#:72196-12) in 24-well, nonpyrogenic, polystyrene plates. Cells were fixed in 4% PFA in PBS at 37 °C for 10 min and washed once in PBS, followed by permeabilization with 0.5% Triton-X in PBS at room temperature for 10 min. The samples were blocked in 3% BSA

in PBS (+0.05% sodium azide) for 1 h at room temperature, incubated with primary antibodies in 3% BSA in PBS (+0.05% sodium azide) for 1 h at room temperature, washed 3× in 3% BSA in PBS (+0.05% sodium azide), followed by secondary antibodies and 4,6-diamidino-2-phenylindole (DAPI) for 1 h at room temperature. Coverslips were then washed 2× in PBS and mounted onto glass slides using ProLong Gold Antifade Reagent (Cell Signaling Technologies, Cat#: 9071). Immunofluorescence was visualized using a Zeiss LSM 710 Confocal microscope (Cell and Developmental Microscopy Core at UPenn) using ZEN v2.3 acquisition software. Images were processed in FIJI (v1.53f51) using equivalent settings.

## 1,6-hexanediol and propylene glycol experiments

To prevent the formation of 52K BMCs in cells, 1,6-hexanediol (Sigma-Aldrich, Cat#: 240117) was diluted in cell culture media at 75 mM (0.8% w/v). Cell culture media was replaced with cell culture media containing 1,6-hexanediol at 8 h post doxycycline induction and treatment was maintained until 24 h post doxycycline induction. At 24 h, cells were washed once in 37 °C PBS and fixed in 4% paraformaldehyde at 37 °C for 10 min before Immunofluorescence analysis.

To disrupt 52K BMCs that had already formed in cells, 1,6-hexanediol was diluted in cell culture media at 400 mM (4.72% w/v). Cell culture media was replaced with cell culture media containing 1,6-hexanediol at 10 h post doxycycline induction and treatment was maintained for 10 min. After 10 min, cells were washed once in 37 °C PBS and fixed in 4% paraformaldehyde at 37 °C for 10 min before immunofluorescence analysis.

Experiments with propylene glycol (Sigma-Aldrich, Cat# P4347, Lot# MKCN1330) were carried out as described above for 1,6-hexanediol, using a concentration of 300 mM (2.19% w/v) to prevent formation of nuclear bodies, and a concentration of 800 mM (5.85% w/v) to disrupt pre-formed nuclear bodies.

To determine the effect of 1,6-hexanediol and propylene glycol on cell membrane integrity and death, A549 cells were incubated in cell culture media supplemented with the indicated concentrations of 1,6-hexanediol for 24 h. After 24 h, cells were trypsinized, resuspended, and stained with 0.4% Trypan Blue. The percentage of cells stained for Trypan Blue (dead cells) was then calculated using a Countess II automated cell counter (Thermo-Fisher, Cat# AMQAX2000) as per the manufacturer's instructions.

To assess the impact of 1,6-hexanediol on condensate formation in vitro, In vitro condensate formation assays were carried out in phase-separation buffer containing 100-, 200-, 300-, 400-, 500-, or 600 mM 1,6-hexanediol or no 1,6-hexanediol as a control.

## Image analysis

Image analysis was carried out using FIJI (v1.53f51). For analysis of 52K BMC size and number per nuclei, 52K BMCs were selected as regions of interest (ROI) and analyzed using the analyze particles and measure functions. ROIs were selected using the Gaussian blur (sigma radius 2), fluorescence intensity thresholding, and binary object selection functions. Nuclei were determined to be antigen-positive if their mean nuclear fluorescence intensity was greater than the mean + two standard deviations of mean nuclear fluorescence intensity in the negative control (uninfected or mock-transfected) nuclei. Cells without BMCs were excluded from

analysis of size and number. For 52K BMC presence or absence, cell nuclei (regions exhibiting DAPI staining) were selected as ROI, and the percentage of cells containing BMCs was quantified. For quantification of viral progeny production via Ad5 DNA Binding Protein immunostaining, see "Viral progeny production". For all image analyses, a minimum of two fields of view were analyzed for each of three independent replicates.

## Fluorescence recovery after photobleaching (FRAP)

Live cell imaging of GFP fusion proteins in transfected HEK-293 cells for FRAP was performed on a Zeiss LSM 880 confocal microscope outfitted with an environmental chamber, an Andor iXon3 EMCCD camera, and a Gataca Systems iLas2 FRAP module operating on MetaMorph acquisition software. All imaging of samples was conducted under standard conditions of 37 °C and 5% $CO_2$. Photobleaching of transiently expressed 52K-GFP (WT, S28/75A, S28/75D) was performed using the 488 nm laser line from an Argon ion laser. Fluorescence was monitored for 2.5 s prior to photobleaching, and recovery was monitored for 22.5 s. For punctate 52K BMCs subjected to whole-BMC bleaching, only BMCs with a diameter less than 1 µm were analyzed. For large 52K BMCs subjected to internal spot bleaching, a fixed laser diameter was used to generate consistent bleach regions with a diameter of 4.155 µm for analysis. Fluorescence recovery was analyzed using FIJI (v1.53f51) using the built-in plugins; create spectrum jru v1, combine all trajectories jru v1, normalize trajectories jru v1, batch FRAP fir jru v1, and average trajectories jru v1. Videos were created using the built-in.AVI file generator in FIJI at the default speed of five frames per second.

## Viral progeny production

Infected cells were harvested by scraping and lysed by four cycles of freeze-thawing in liquid nitrogen and a 37 °C water bath. Cells were harvested at 48 hpi unless stated otherwise. Cell debris was removed from lysates by centrifugation at max speed, 4 °C, 5 min. For analysis of virus yield, lysates were diluted serially in DMEM supplemented with 2% FBS and 1% Pen/Strep and used to infect A549 cells. Infection media was removed 2 h after infection, cells were washed 1× in PBS to remove excess virus, and cells were overlaid with growth media. Cells were incubated for 24 h prior to fixation in 4% paraformaldehyde and analyzed by immunofluorescence–confocal microscopy using immunostaining of the viral DNA binding protein (DBP) as an indicator of infection. For three independent replicates, two fields of view were captured and the percentage of DBP-positive cells was determined. The serial dilution resulting in closest to 50% DBP-positive cells was selected for the calculation of progeny production. The total cell number was determined by counting cells grown in parallel under equivalent conditions. The number of focus forming units (FFU) was calculated as the product of total cell number and % of antigen-positive cells, adjusting for the Poisson distribution. Virus input was determined by harvesting infected cells at 4 hpi, the number of infectious units was determined, and the mean of replicates was calculated.

## Virus purification and analysis of packaged or incomplete viral particles

For isolation of packaged or incomplete viral particles, large-scale lysates were generated from 10 × 15 cm tissue culture dishes of

confluent cells (~600 million cells). Cells were washed 1× in PBS, harvested by scraping, pooled, and pelleted by centrifugation at 1500 rpm for 5 min. Cell pellets were lysed by four cycles of freeze-thawing in liquid nitrogen and a 37 °C water bath, and cell debris was removed by centrifugation at max speed, 4 °C, 5 min. PBS was added to lysates to a final volume of 7 mL. To prepare cesium chloride (CsCl) density gradients, 2.5 mL 1.34 g/cc CsCl was added to 13.2 mL Ultra-Clear ultracentrifuge tubes (Beckman-Coulter, Cat#: 344059) and carefully underlaid with 2.5 mL 1.43 g/cc CsCl. Clarified viral lysates were gently layered on top. Loaded samples were centrifuged at 25,000 rpm for 2 h at 4 °C using an Optima XPN-80 ultracentrifuge (Beckman-Coulter), and SW-41 Ti rotor (Beckman-Coulter). Tubes were removed from the ultracentrifuge and placed in a suspended laboratory clamp for imaging and harvesting of AdV particle bands. Both unpackaged and packaged bands were harvested in a sterile 15-mL conical tube, resuspended in 7 mL of sterile PBS, and layered on top of a fresh CsCl density gradient. Tubes were spun a second time in the ultracentrifuge at 25,000 rpm for 16 h at 4 °C to achieve optimal separation of packaged and unpackaged particle bands. Tubes were then removed from the ultracentrifuge and placed in a tube rack for final imaging.

To quantify viral particles, viral bands were fractionated and collected by dripping through a puncture introduced in the bottom of the tube using an 18-gauge syringe needle. The volume of each fraction was measured, and virus was serially diluted in 0.1% SDS in PBS. Viral particles were disrupted by incubation at 56 °C, 1000 RPM for 30 min in a thermomixer, followed by boiling at 95 °C for 5 min. Each dilution (1 μL) was then used for measuring A260 and A280, and viral particle counts were calculated from these absorbance values using a previously published method (Maizel et al, 1968).

### Reproducibility and statistics

All experimental observations were confirmed by independent repeat experiments, with the exception of the in vitro kinase screen, which was performed once. No blinding or sample randomization was employed during data capture or analysis. Where possible, matched experiments were performed in parallel. No data were excluded from data analysis. No statistical tests were performed to pre-determine sample size. Where data from independent repeats are pooled, the experimental outcome was confirmed to be consistent between repeats. Unless stated otherwise, the mean of numerical data is shown, with error bars representing the standard deviation of the sample. Statistics were performed using GraphPad Prism version 9. All $t$ tests are two-sided. Data was tested for normality before applying either parametric or non-parametric statistical tests, where $n = 3$, a normal distribution and equal variance were assumed when determining statistical tests.

## Data availability

All graphed numerical data, unprocessed micrographs, as well as uncropped immunoblots blots, and Coomassie gels corresponding to main figures are available as source data. Unprocessed microscopy images will be made available on request. The 52K immunoprecipitation mass spectrometry proteomics data have been deposited to the ProteomeXchange Consortium via the PRIDE

(Perez-Riverol et al, 2022) partner repository with the dataset identifier PXD046971. Readers may access the dataset by navigating to PRIDE via the link: https://www.ebi.ac.uk/pride/archive/projects/PXD046971. Where available, materials generated as part of this study will be made available upon request.

## Peer review information

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

## Acknowledgements

We thank members of the Weitzman Lab for insightful discussions and input. We are grateful to P Hearing, M Imperiale, A Levine, D Ornelles, and J Wilson for generous gifts of reagents. We thank Dr. Steven Pelech for his expertise and advice on kinase biology in the context of virus infection. We thank Paul Lieberman, Marisa Bartolomei, Rahul Kohli, and Fange Liu for their helpful insight regarding project direction and experimental design. We thank Amber Abbott, Joseph Dybas and Namrata Kumar for their thoughtful feedback and careful reading of the manuscript. We thank the UPenn Cell and Developmental Biology Microscopy Core for imaging assistance. We thank Kushol Gupta of the UPenn Johnson Foundation Biophysical and Structural Biology Core for his insight and guidance. This research was supported by NIAID grants R01-AI145266 (MDW), R01-AI121321 (MDW), R01-AI118891 (MDW and BAG), and NIH grant T32-GM007229 (NG/UPenn Cell and Molecular Biology Department).

## Author contributions

**Nicholas Grams**: Conceptualization; Data curation; Formal analysis; Validation; Investigation; Visualization; Methodology; Writing—original draft; Writing—review and editing. **Matthew Charman**: Conceptualization; Data curation; Formal analysis; Supervision; Validation; Investigation; Visualization; Methodology; Writing—original draft; Writing—review and editing. **Edwin Halko**: Resources; Investigation; Methodology; Writing—review and editing. **Richard Lauman**: Data curation; Formal analysis; Investigation; Methodology; Writing—review and editing. **Benjamin A Garcia**: Funding acquisition; Project administration. **Matthew D Weitzman**: Supervision; Funding acquisition; Project administration; Writing—review and editing.

## Disclosure and competing interests statement

The authors declare no competing interests.

# Expanded View Figures

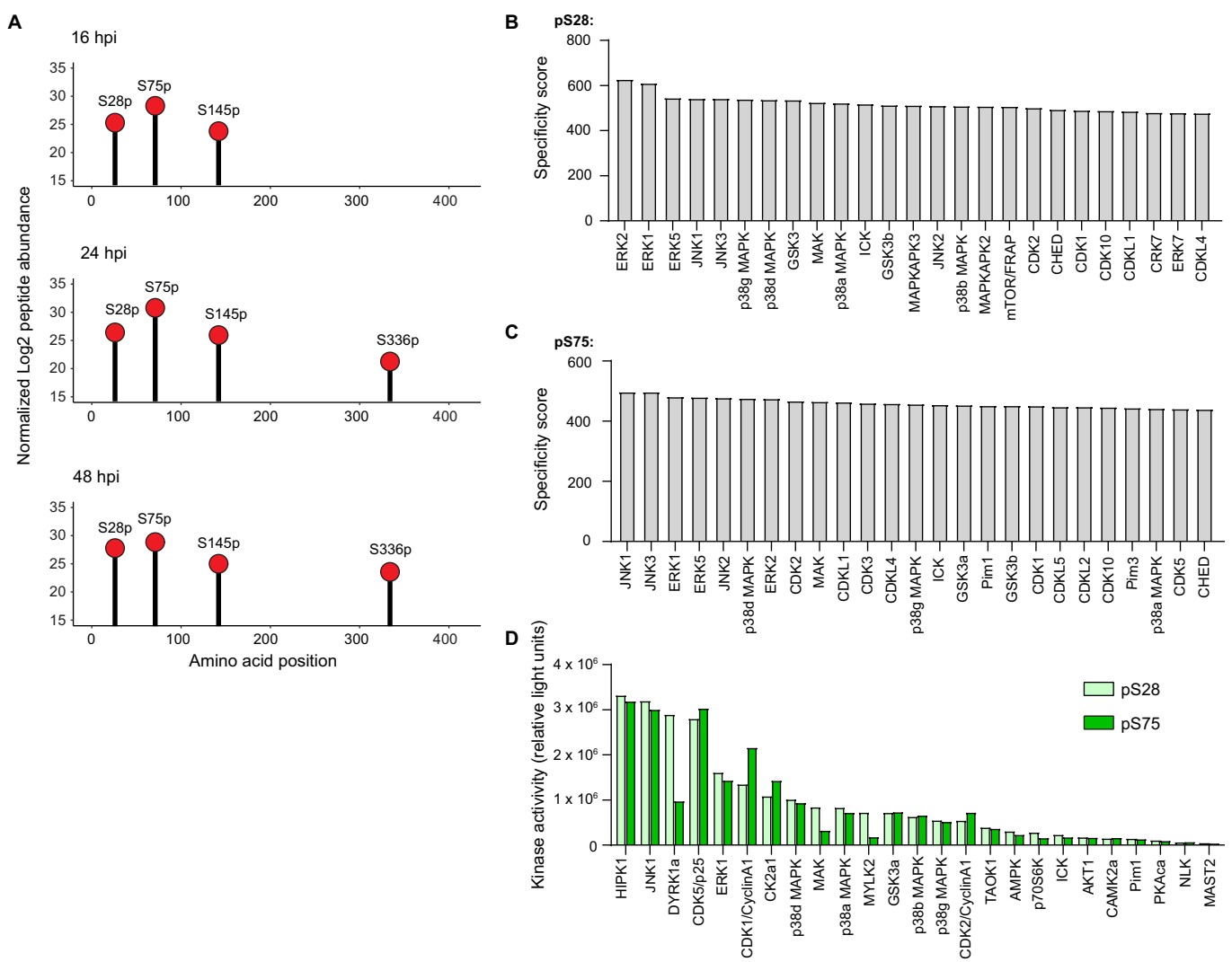

**Figure EV1.   Site specific phosphorylation of 52K.**

(A) Analysis of posttranslational modification-mass spectrometry dataset showing site specificity of phosphorylation during adenovirus (AdV) infection of A549 cells at 16-, 24-, or 48-h post infection (hpi). Red lollipops denote phosphorylated serine residues S28, S75, S145, and S336 respectively. Modifications shown as log2 abundance of modified peptide relative to total peptide. (B) Bar graph showing the top 25 human kinases with the highest in silico catalytic site specificity scores matching the 52K protein S28 phospho-site. (C) Bar graph showing the top 25 human kinases with the highest in silico catalytic site specificity scores matching the 52K protein S75 phospho-site. (D) Bar graph showing the relative activity of 25 human kinases in vitro when provided with peptide substrates matching either the 52K S28 or S75 phospho-sites. Kinase activity was measured as relative light units.

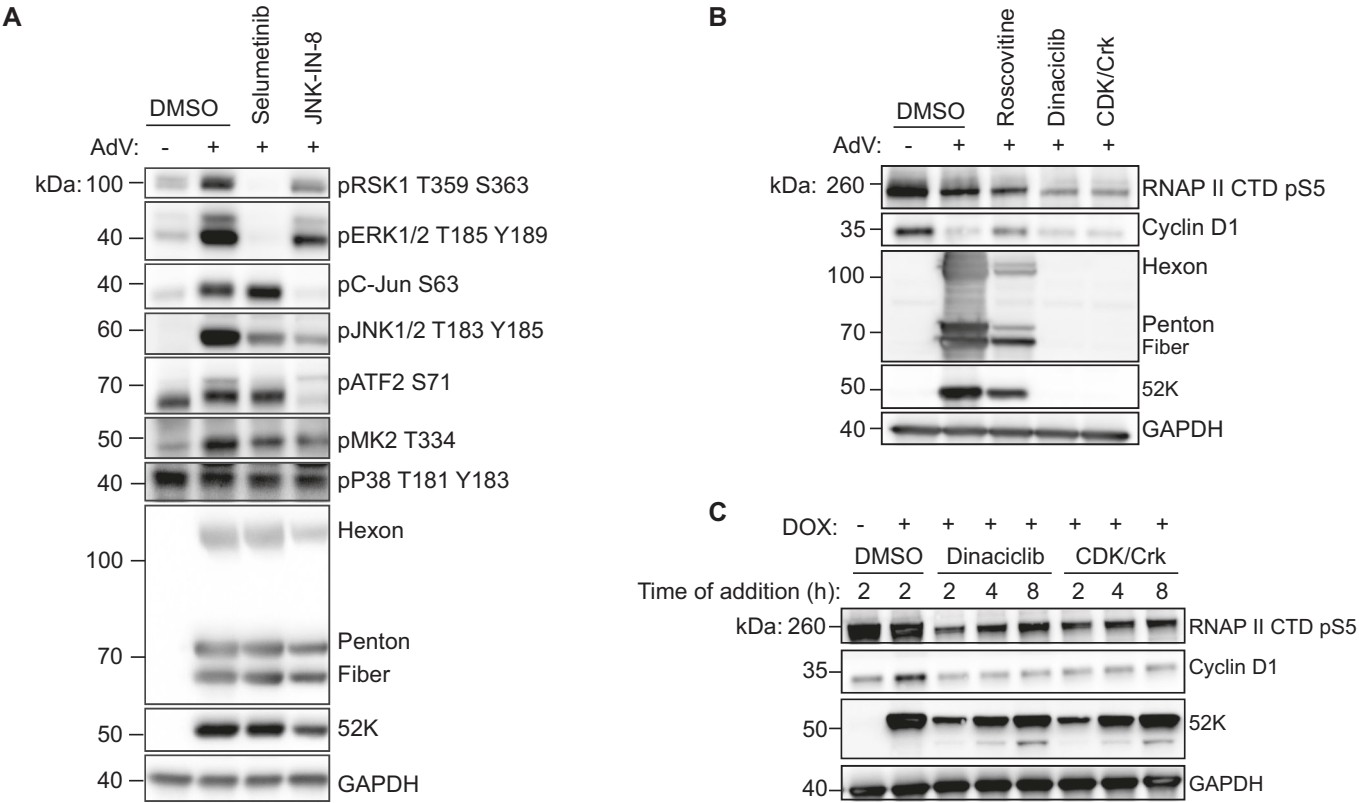

**Figure EV2. Inhibition of MAPK and CDK family kinases.**

(A) Immunoblot showing 52K and viral late proteins (Hexon, Penton, Fiber) from adenovirus (AdV) infected A549 cells treated with MAP Kinase inhibitors. DMSO control or 10 μM of kinase inhibitor (Selumetinib = MEK1/2, JNK-IN-8 = JNK1/2) was added at 2 h post infection (hpi). Infection was allowed to progress in the presence of inhibitor or DMSO for 24 h before being harvested for immunoblot. Downstream-target controls (pRSK1, pERK1/2 = Selumetinib. pC-Jun, pJNK1/2, pATF2 = JNK-IN-8, pMK2) confirm efficacy of the inhibitors. Phosphorylated P38 (pP38) serves as a kinase specificity control. GAPDH is included as a loading control. Representative of $n = 3$ independent replicates. (B) Immunoblot showing 52K and viral late proteins (Hexon, Penton, Fiber) from AdV-infected A549 cells treated with Cyclin-Dependent Kinase inhibitors. DMSO control or 10 μM of kinase inhibitor (Roscovitine = CDK2/5, Dinaciclib = CDK4/6, CDK/crk = CDK1/2/4/5/7/9) was added at 2 hpi. Infection was allowed to progress in the presence of inhibitor or DMSO for 24 h before being harvested for immunoblot. Downstream-target controls (RNAP II CTD pS5 and Cyclin D1) confirm efficacy of the inhibitors. GAPDH is included as a loading control. Representative of $n = 3$ independent replicates. (C) Immunoblot showing 52K from transgenic A549 cells treated with Cyclin-Dependent Kinase inhibitors. Expression of WT 52K was induced by addition of doxycycline and DMSO control or 10 μM of kinase inhibitor (Dinaciclib = CDK4/6, CDK/crk = CDK1/2/4/5/7/9) was added at the indicated time post induction. Protein accumulation was allowed to progress in the presence of inhibitor or DMSO for 24 h before being harvested for immunoblot. Downstream-target controls (pRNAP II CTD pS5 and Cyclin D1) confirm efficacy of the inhibitors. GAPDH is included as a loading control. Representative of $n = 3$ independent replicates.

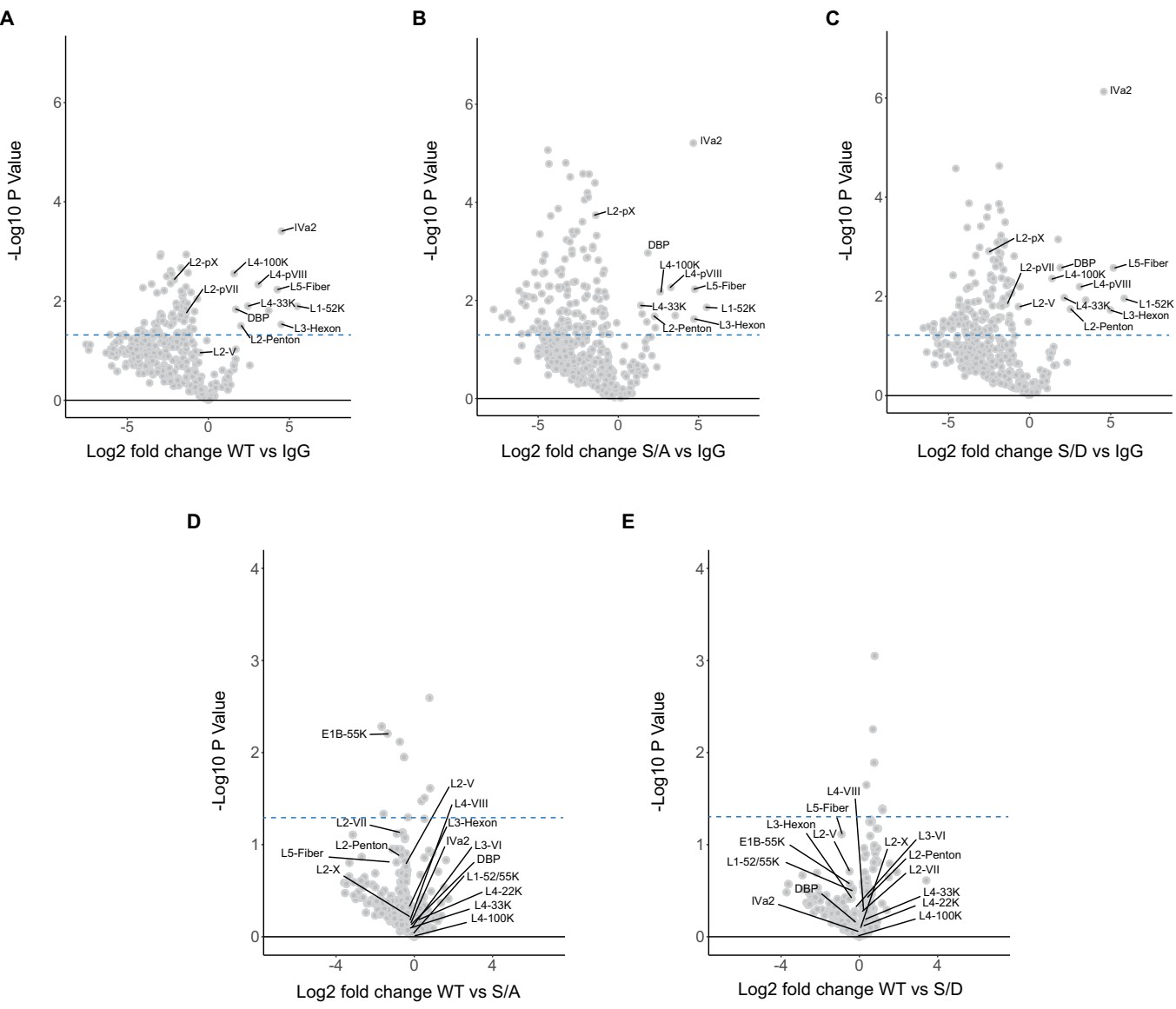

**Figure EV3.  Phosphorylation of 52K does not influence interactions with viral proteins.**

(**A**) Volcano plot showing mean relative abundance of proteins co-precipitated from Δ52K mutant adenovirus (AdV) infected HEK-293 cell lysates with wild-type 52K (WT) versus IgG control. Viral proteins detected are labeled. *n* = 4 independent replicates. (**B**) Volcano plot showing mean relative abundance of proteins co-precipitated from Δ52K mutant AdV-infected HEK-293 cell lysates with S28/75 A (S/A) mutant versus IgG control. Viral proteins detected are labeled. *n* = 4 independent replicates. (**C**) Volcano plot showing mean relative abundance of proteins co-precipitated from Δ52K mutant AdV-infected HEK-293 cell lysates with S28/75D (S/D) mutant versus IgG control. Viral proteins detected are labeled. *n* = 4 independent replicates. (**D**) Volcano plot showing mean relative abundance of proteins co-precipitated from Δ52K mutant AdV-infected HEK-293 cell lysates with wild-type 52K (WT) compared to the phosphorylation-deficient S/A mutant. Viral proteins detected are labeled. *n* = 4 independent replicates. (**E**) Volcano plot showing mean relative abundance of proteins co-precipitated from Δ52K mutant AdV-infected HEK-293 cell lysates with wild-type 52K (WT) compared to the phosphorylation-mimetic S/D mutant. Viral proteins detected are labeled. *n* = 4 independent replicates. Data information: Data are presented as the average of four independent replicates. For all sample comparisons, two-sided Student's *T* tests were used to determine significant changes with a *P* value < 0.05. The dotted blue line shows the statistical cutoff of *P* < 0.05 (**A–E**).

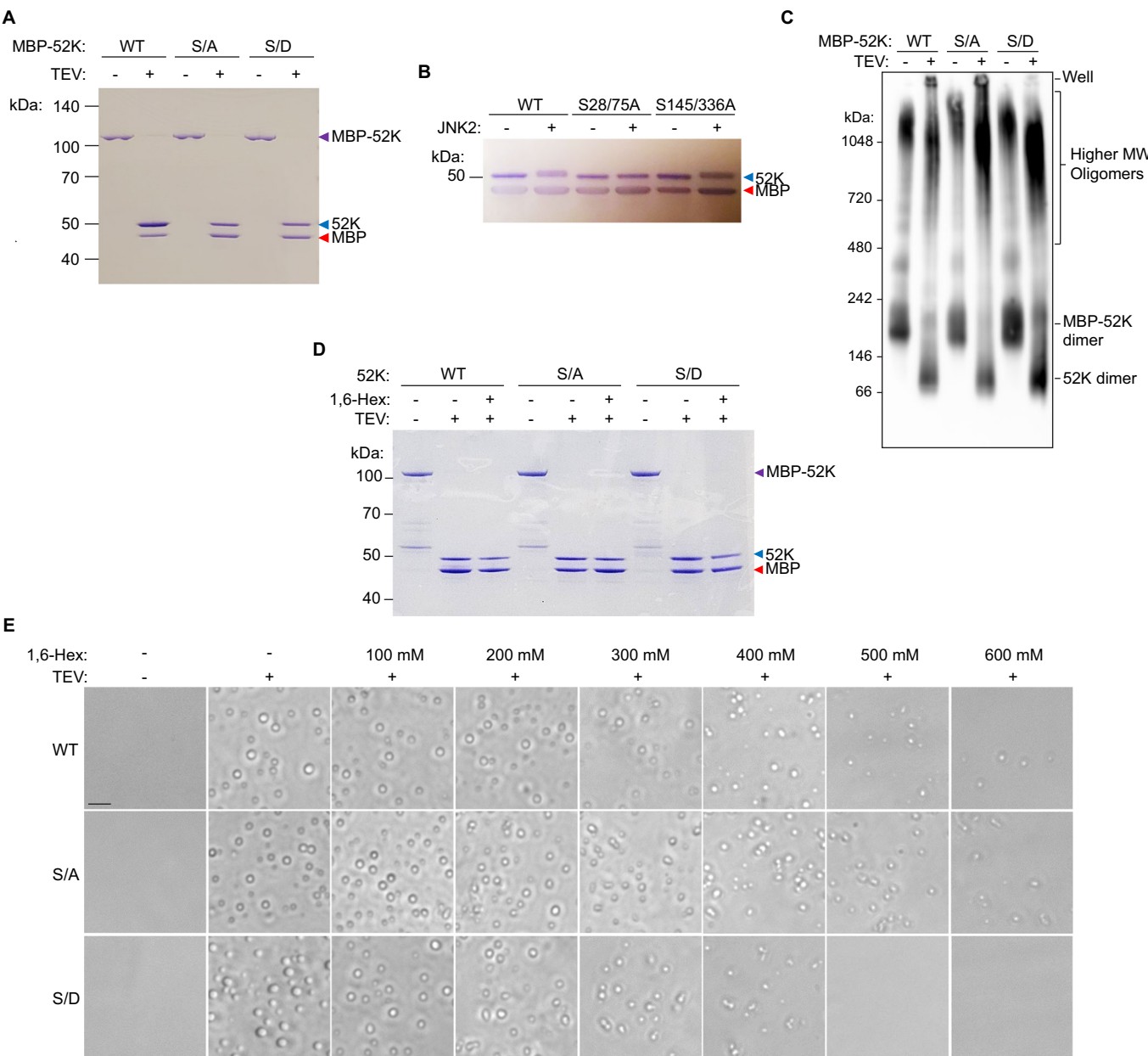

**Figure EV4. Impact of phosphorylation on the 52K protein in vitro.**

(A) Coomassie Brilliant Blue stained SDS-PAGE showing maltose binding protein (MBP) tagged wild-type (WT), S28/75 A (S/A), or S28/75D (S/D) 52K fusion proteins with (+) or without (−) Tobacco Etch Virus (TEV) protease cleavage for 1 h at room temperature. Representative of $n = 3$ independent replicates. (B) Coomassie Brilliant Blue stained SDS-PAGE showing electrophoretic mobility of wild-type 52K (WT), S28/75 A or S145/336 following treatment with (+) or without (−) JNK2 kinase. The MBP tag was removed by TEV protease cleavage. Representative of $n = 3$ independent replicates. (C) Immunoblot for 52K following native-PAGE of MBP-tagged wild type (WT), S28/75 A, or S28/75D 52K fusion proteins with (+) or without (−) TEV protease cleavage for 1 h at room temperature. Representative of $n = 3$ independent replicates. (D) Coomassie Brilliant Blue stained SDS-PAGE showing maltose binding protein (MBP) tagged wild type (WT), S28/75 A (S/A), or S28/75D (S/D) 52K fusion proteins with (+) or without (−) Tobacco Etch Virus (TEV) protease cleavage for 1 h at room temperature in the absence (−) or presence of 600 mM 1,6-hexanediol (+). Representative of $n = 3$ independent replicates. (E) Wide-field microscope images showing condensates formed by phase separation of 5 μM wild-type 52K (WT), S28/75 A (S/A), or S28/75D (S/D) in vitro in the absence (−) or presence of 1,6-hexanediol (+) at the indicated concentrations (100 mM–600 mM). Phase separation was enabled by removal of the MBP tag by addition of TEV protease (+). A no TEV control (−) is included. Scale bars = 10 μm. Representative of $n = 3$ independent replicates.

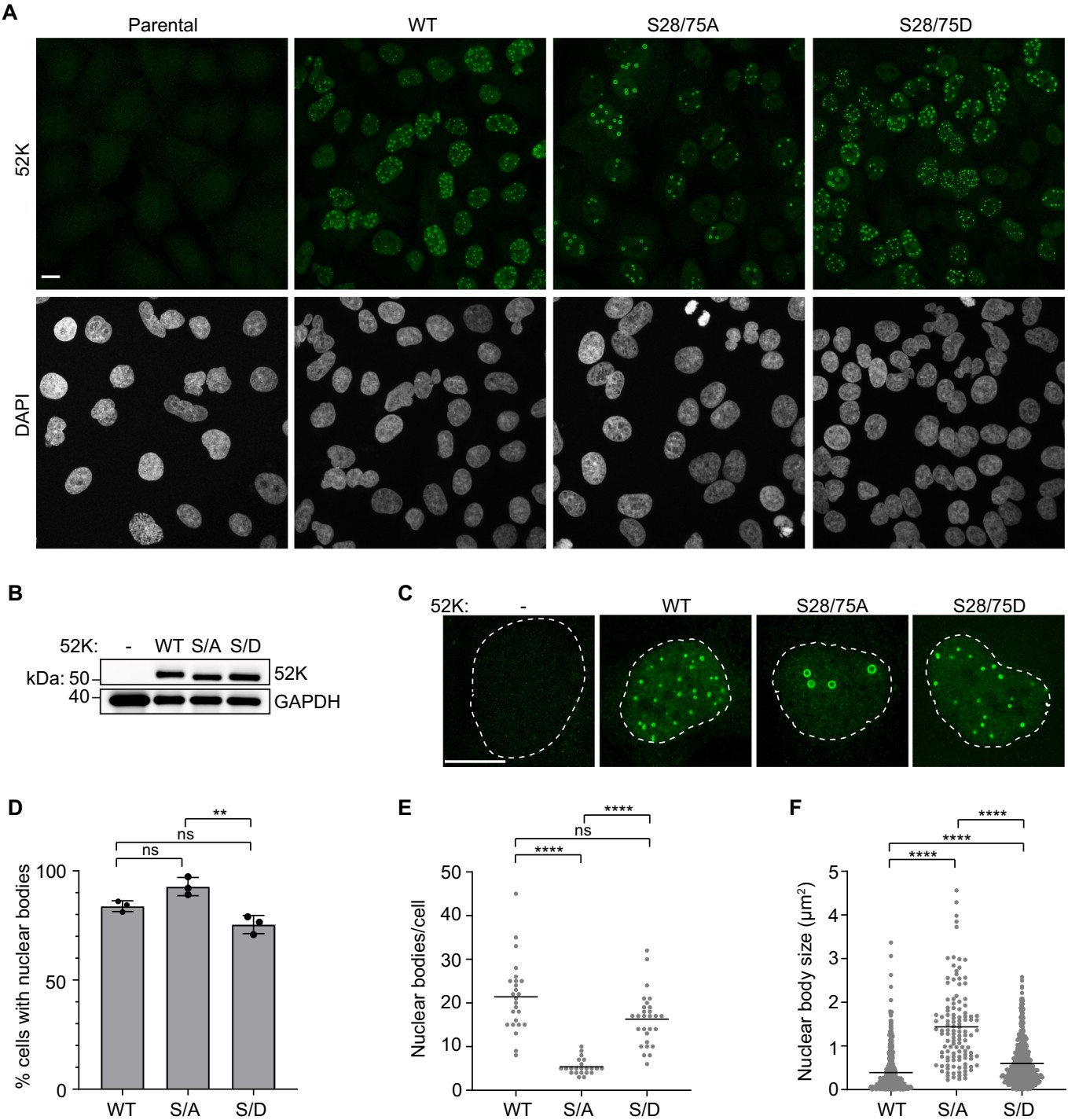

**Figure EV5. Phosphorylation of the 52K protein IDR alters nuclear body morphology in HEK-293 cells.**

(A) Immunofluorescence–confocal microscope images showing cell to cell variability in size and number of nuclear bodies in parental or transgenic A549 cell lines expressing wild-type 52K (WT), or phosphorylation mutants S28/75 A, or S28/75D at 24 h after induction of expression by addition of doxycycline. Scale bar = 5 μm. Representative of $n = 3$ independent replicates. (B–F) HEK-293 cells expressing wild-type 52K (WT), phosphorylation mutant S28/75A (S/A) or S28/75D (S/D), or mock-transfected control (−) as indicated. (B) Immunoblot showing the levels of ectopically expressed 52K. GAPDH is shown as a loading control. Representative of $n = 3$ independent replicates. (C) Immunofluorescence–confocal microscope images showing the morphology of 52K nuclear bodies. Nuclei outlined (dashed white line). Scale bar = 5 μm. Representative of $n = 3$ independent replicates. (D) Percentage of cells with nuclear bodies. $n = 3$ independent replicates each consisting of 3 fields of view analyzed. (E) Number of nuclear bodies per cell. $n = 3$ independent replicates pooled. A total of 25 (WT), 22 (S/A), or 28 (S/D) cells were analyzed. (F) Nuclear body size. $n = 3$ independent replicates pooled. Data information: Data are presented as mean ± standard deviation (D–F). One-way ANOVA with Tukey's pairwise comparison tests (D) or Kruskal–Wallis ANOVA with Dunn's pairwise comparison tests (E, F). NS $P > 0.05$, **$P < 0.01$, ****$P < 0.0001$.

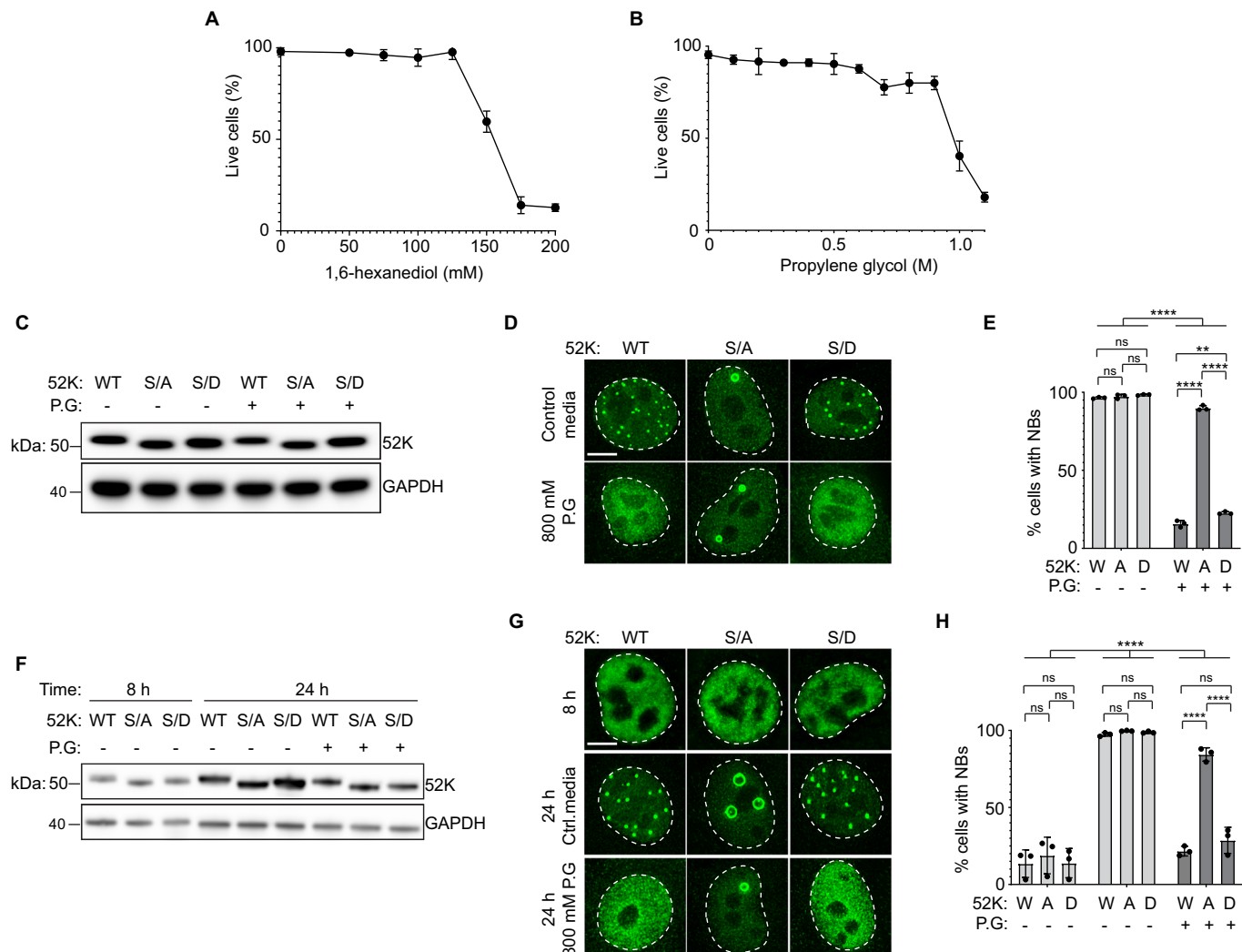

**Figure EV6. Phosphorylation of the 52K IDR impacts sensitivity of nuclear bodies to diols.**

(**A**) Percentage of live cells after incubation of A549 cells in the presence of the indicated concentration of 1,6-hexanediol for 24 h as determined by trypan blue staining. $n = 3$ independent replicates. (**B**) Percentage of live cells after incubation of A549 cells in the presence of the indicated concentration of propylene glycol for 24 h as determined by trypan blue staining. $n = 3$ independent replicates. (**C–E**) Expression of the 52K protein and nuclear body formation in transgenic A549 cells. Expression of wild-type 52K (WT), S28/75 A (S/A), or S28/75D (S/D) was induced by addition of doxycycline for 10 h before addition of 800 mM propylene glycol (P.G) for 10 min prior to analysis. $n = 3$ independent replicates. (**C**) Immunoblot showing levels of 52K. GAPDH is shown as a loading control. (**D**) Representative confocal microscope images showing the localization of 52K. Nuclei are outlined (dashed white line). Scale bar = 5 μm. (**E**) Percentage of cells with nuclear bodies. Each independent replicate consisted of 3 fields of view. (**F–H**) Transgenic A549 cells. Expression of wild-type 52K (WT), S28/75 A (S/A), or S28/75D (S/D) was induced by addition of doxycycline for 8 h before incubation with control media or media containing 300 mM propylene glycol (P.G) for an additional 16 h (24 h post induction). $n = 3$ independent replicates. (**F**) Immunoblot showing levels of 52K. High exposure panel shows the lower levels of expression at 8 h. GAPDH is shown as a loading control. (**G**) Representative images showing the localization of 52K. Nuclei are outlined (dashed white line). Scale bar = 5 μm. (**H**) Percentage of cells with nuclear bodies. Each independent replicate consisted of 3 fields of view. Data information: Data are presented as mean ± standard deviation (**A, B, E, H**). Two-way ANOVA with Tukey's multiple comparison's tests (**E, H**). NS $P > 0.05$, **$P < 0.01$, ****$P < 0.0001$.

