## [Peer Review File · The EMBO Journal]

Phosphorylation regulates viral biomolecular condensates to promote infectious progeny production

Matthew Weitzman, Matthew Charman, Nicholas Grams, Edwin Halko, Richard Lauman, and Benjamin Garcia
DOI: 10.15252/emboj.2023115032

Corresponding author(s): Matthew Weitzman (weitzmanm@email.chop.edu) , Matthew Charman (charmanm@chop.edu)

Review Timeline:

Submission Date:	18th Jul 23
Editorial Decision:	17th Aug 23
Revision Received:	15th Nov 23
Editorial Decision:	4th Dec 23
Revision Received:	6th Dec 23
Accepted:	7th Dec 23

Editor: Hartmut Vodermaier

Transaction Report:

Prof. Matthew D Weitzman
The Children's Hospital of Philadelphia
Pathology & Laboratory Medicine
4050 Colket Translational Research Building
3501 Civic Center Blvd
Philadelphia, PA 19104

17th Aug 2023

Re: EMBOJ-2023-115032
Phosphorylation regulates viral biomolecular condensates to promote infectious progeny production

Dear Matt,

Thank you for submitting your study on phosphorylation regulation of biomolecular condensates involved in adenovirus packaging for our editorial consideration. It has now been assessed by two expert referees, whose reports are copied below. As you will see, both reviewers appreciate the potential interest and timeliness of the work, but there are also a number of substantive concerns that would need to be adequately addressed before publication. In particular, referee 1 raises a key caveat regarding the use of 1,6-hexanediol, the interpretation of subtle differences in packaging, and wants deeper exploration of direct/macrosopic functional consequences of the analyzed phosphorylations. Furthermore, both referees have related questions regarding the effects on oligomerization/multivalent protein interactions and biophysical properties.

Should you be able to satisfactorily address these key issues, and to respond to the other more specific queries, we would be interested in pursuing a revised manuscript further for publication. Since it is our policy to consider only a single round of major revision and therefore important to fully answer to all comments at the time of resubmission, I would however invite you to get back to me with a tentative response letter/revision plan already during the early stages of the revision work. On the basis of this response, we could then further discuss the revision requirements and how to best address the key concerns e.g. via a follow-up video call. I should add that we could also offer extension of the default three-months revision period if needed, with our 'scooping protection' (meaning that competing work appearing elsewhere in the meantime will not affect our considerations of your study) remaining of course valid also throughout this extension.

Detailed information on preparing, formatting and uploading a revised manuscript can be found below and in our Guide to Authors. Thank you again for the opportunity to consider this work for The EMBO Journal, and I look forward to hearing from you in due time.

With kind regards,

Hartmut

9) Digital image enhancement is acceptable practice, as long as it accurately represents the original data and conforms to community standards. If a figure has been subjected to significant electronic manipulation, this must be clearly noted in the figure legend and/or the 'Materials and Methods' section. The editors reserve the right to request original versions of figures and the original images that were used to assemble the figure. Finally, we generally encourage uploading of numerical as well as gel/blot image source data; for details see: embopress.org/page/journal/14602075/authorguide#sourcedata

At EMBO Press, we ask authors to provide source data for the main manuscript figures. Our source data coordinator will contact you to discuss which figure panels we would need source data for and will also provide you with helpful tips on how to upload and organize the files.

In the interest of ensuring the conceptual advance provided by the work, we recommend submitting a revision within 3 months (15th Nov 2023). Please discuss the revision progress ahead of this time with the editor if you require more time to complete the revisions. Use the link below to submit your revision:

Link Not Available

Referee #1:

The manuscript by Grams and colleagues presents an interesting examination of the role phosphorylation plays in modulating viral biomolecular condensates, specifically focusing on the 52K protein of adenovirus. The investigation into the regulation of these condensates during adenovirus infection and the subsequent influence on viral progeny production is both timely and topical. The authors provide evidence suggesting that specific phosphorylation events on the 52K protein modulate the nature of these condensates, which in turn impacts the production of infectious viral particles. Interestingly, similar findings have been reported for other viruses, notably, rotaviruses, for which various studies, including reverse genetics studies, have shown that the phosphorylation of the condensate-forming scaffold protein NSP5 is important, but not essential for viral replication and progeny formation (PMCID: PMC6912106; 2019; PMC8561643; 2021), further supporting the generality of the scaffold protein phosphorylation and their roles in viral replication and genome packaging.

Major Concerns:

1. Functional consequences of condensate formation: While the paper does discuss the role of phosphorylation in the modulation of condensates, it would be helpful to have a deeper exploration into the direct functional consequences of these modifications. How do these changes at the molecular level translate to the macroscopic properties of the condensates? The authors refer to these condensates as 'coordinating viral genome packaging and capsid assembly' but so far the data only support the role of 52K in genome packaging, while condensate formation may be unrelated to the protein's role in capsid

assembly. It would be also helpful to have a deeper exploration into the direct functional consequences of these modifications. How do these changes at the molecular level (e.g., structure/(dis)order transitions of 52K, its oligomerization etc) translate to the macroscopic properties of the condensates?)

2. The study shows that serine residues 28 and 75 within the N-terminal IDR of 52K are crucial for the modulation of viral condensates. However, more evidence is needed to demonstrate the specificity of these phosphorylation events given that the ADP-Glo assay was carried out with peptides. Are these the only residues that play this role, or are there other phosphorylation sites that have not been studied? Moreover, can the authors comment on the localization and activity of the identified top-ranked kinases shown in Fig EV1C - presumably, these should be active in the nucleus to phosphorylate the 52K where it forms inclusions? Lastly, if the authors know which kinases are essential for 52K phosphorylation, can they identify suitable kinase inhibitors, even low-specificity ones, which could be used to enhance their studies? Lastly, *in vivo* their analysis was carried out at 24 hpi (Line 157) - it is likely that these sites may dynamically change throughout the infection cycle and will vary with the multiplicity of infection. Did the authors look at the sites at earlier points of infection, and varying MOI.

3. Can the authors provide an explanation for why the observed *in vitro* droplets coalesce better when phosphorylated (larger and fewer droplets, line 216), while in cells they observe the opposite effects (smaller and more numerous when phosphorylated, line 236).

4. Fig.3 - the use of 1,6 hexanediol. Although I appreciate the usefulness of 1,6HD in such studies, I would be extremely careful in interpreting these results, particularly given the inhibitory effects of this diol on cellular kinases that are probably involved in 52K phosphorylation (PMCID: PMC7948595; 2021). The authors could try propylene glycol, which has been used in viral studies (PMCID: PMC8561643; 2021) and appear to be much better tolerated by cells over extended periods of time. Importantly, can the authors demonstrate that *in vitro* 75 mM 1,6HD does alter the phase behavior of 52K condensates (I really doubt that it does at this low concentration given multiple studies of this alcohol and how it modulates condensates). If this is the case, I doubt that the observed effect of 75 mM hexanediol has anything to do with its disruptive effect on condensates, and it has more something to do with its toxicity and inhibitory effects on cells.

5. FRAP studies: the reported slower recovery rates - are the authors confident that these differences are directly attributed to the phosphorylation status or differences in the molecular composition of such inclusions? Do they know if lower/higher amounts of e.g., DNA in distinct puncta may also contribute to slower/faster diffusion? Do they know if phosphorylation may alter the oligomerization of their proteins, which would in turn change the diffusion and recovery rates?

6. The results of virus validation studies are also intriguing: the authors show that although S28/75A mutant can contribute to infectious progeny production, this ability is attenuated compared to phosphomimetics. How comparable are the phosphomimetic and the WT viral production in this case? Are they functionally interchangeable?

7. Lastly, the important conclusion of the paper is the notion that 52K phosphorylation is involved in AdV genome packaging. What I find intriguing in Fig5. is that the amount of viral DNA produced when the mutant virus replicates is most likely lower compared to the WT virus (the authors should provide further data to compare DNA synthesis in all viruses and mutants examined). This is important for their interpretation, as the lower packaging efficiency could be simply attributed to lower DNA synthesis, or slower DNA replication kinetics, in which case more empty particles could be formed. I appreciate studies of genome packaging efficiencies are tricky and hard to quantify, but a lot of caution must be taken when interpreting such subtle differences in packaging, as shown in Fig.5. Additionally, the authors interpret packaging differences as an attempt to generate failed assembly intermediates enriched for 52K (Line 348) - can they also analyze their viral preparations (shown in Fig.5E) by mass-spectrometry to see if there are any differences in protein abundances?

Minor:

1. Introduction: The authors may consider including more references on the role and functional consequences of scaffold phosphorylation in the formation of viral inclusions, and genome replication and packaging.

2. Figure 1. AlphaFold predictions - given that AlphaFold is not very good at assessing something which is supposed to lack the structure (IDPs), I suggest expressing the regions of interest and the 52K and checking their secondary structure content by circular dichroism spectroscopy. Similar experiments with phosphomimetics would also show if there are any structural rearrangements or ordering of IDRs upon phosphorylation.

3. Lines 262&265: can the authors provide errors for the reported percentage numbers of cells analyzed?

4. Same for reported FRAP recovery rates, can the authors provide the errors, please?

Referee #2:

The author previously reported that human Adenovirus (AdV) virus assembly takes place in phase separated compartments in the nuclei of infected cells, termed biomolecular condensates (BMCs). The assembly of infectious AdV requires a number of characterized viral genes products, including the L1-52K protein investigated in this study. These virus assembly proteins interact with a specific packaging domain in the viral genome and/or with each other and virus capsid proteins.

In this report, the authors further characterize the role of the AdV L1-52K protein previously shown to nucleate the formation of BMCs. The L1-52K protein was previously shown to be phosphorylated at several specific sites. The authors validate these findings and show that two sites of phosphorylation are located in the disordered N-terminal region of the protein. They make and analyze two mutant proteins: one changes the two phosphorylated serine residues to alanines, and the second replaces the serines with aspartic acid residues as phospho-mimetics. In a convincing series of experiments, the authors show that phosphorylation of serine residues 28 and 75 within the N-terminal disordered region of L1-52K protein regulates viral

condensates in vitro and in vivo. The phospho-mimetic mutant functions like wild-type, whereas the alanine substitution mutant displays altered properties of solubility and mobility in vivo. These phenotypes parallel results examining virus assembly and infectious virus production. The authors conclude that L1-52K phosphorylation alters the physical properties of the protein to promote proper BMC assembly and recruitment of AdV virus assembly and DNA packaging machinery.

This a thorough study with well designed experiments and convincing results. The manuscript is written for a broad audience. The findings are a bit specialized and perhaps of greater interest to virologists than the general readership. But the results are important and extend to a significant extent the author's recent report about phase separated AdV assembly compartments in the nucleus.

I have no major concerns about the manuscript. The authors suggest that L1-52K protein phosphorylation may stabilize multivalent interactions of this protein. Do the authors have any in vitro biophysical data to support this idea? This would greatly strengthen the manuscript, although the studies as presented stand alone of their own merit.

Minor points:

1. The standard nomenclature in the AdV field is to indicate the coding region along with the molecular weight. For clarity, the authors should call the 52K protein L1-52K. Similarly call 100K L4-100K, etc. This should be done to avoid any confusion (eg. E1B-55K vs. L1-52/55K, called 52K here).
2. Fig. 1J. Why is the DBP coding region enriched in the ChIP analysis? This does not make sense sine the packaging domain is at the left end of the viral genome.
3. Lines 331-332: mature AdV bands at 1.34 g/cc and empty particles band at 1.29 g/cc (not 1.43 and 1.34 as stated).

Referee #1:

The manuscript by Grams and colleagues presents an interesting examination of the role phosphorylation plays in modulating viral biomolecular condensates, specifically focusing on the 52K protein of adenovirus. The investigation into the regulation of these condensates during adenovirus infection and the subsequent influence on viral progeny production is both timely and topical. The authors provide evidence suggesting that specific phosphorylation events on the 52K protein modulate the nature of these condensates, which in turn impacts the production of infectious viral particles. Interestingly, similar findings have been reported for other viruses, notably, rotaviruses, for which various studies, including reverse genetics studies, have shown that the phosphorylation of the condensate-forming scaffold protein NSP5 is important, but not essential for viral replication and progeny formation (PMIDs: PMC6912106; 2019; PMC8561643; 2021), further supporting the generality of the scaffold protein phosphorylation and their roles in viral replication and genome packaging.

Major Concerns:

1. Functional consequences of condensate formation: While the paper does discuss the role of phosphorylation in the modulation of condensates, it would be helpful to have a deeper exploration into the direct functional consequences of these modifications. How do these changes at the molecular level translate to the macroscopic properties of the condensates? The authors refer to these condensates as 'coordinating viral genome packaging and capsid assembly' but so far the data only support the role of 52K in genome packaging, while condensate formation may be unrelated to the protein's role in capsid assembly. It would be also helpful to have a deeper exploration into the direct functional consequences of these modifications. How do these changes at the molecular level (e.g., structure/(dis)order transitions of 52K, its oligomerization etc.) translate to the macroscopic properties of the condensates?).

Our previous work (PMID: 37020020) indicates that viral condensates formed by the 52K protein function to regulate viral assembly, such that capsid assembly is coordinated with the provision of viral genomes required to produce complete packaged particles (rather than unpackaged dead-end products that result when these processes are not coordinated). The data presented in our current submission indicates that phosphorylation of 52K promotes this coordinating function, thus promotes assembly of particles that contain genomes. To be clear, we know that condensates are not essential for capsid assembly, as empty capsids form in the absence of 52K (PMID: 37020020, Figure 5 E & F).

However, given our proposed model, it seems possible that the organization of viral capsid proteins into viral condensates would allow for greater total particle formation by maintaining high concentrations of these proteins in a state that remains conducive to assembly (rather than some kind of deleterious aggregation). Our data our consistent with the notion that functional condensates also promote the total number of capsids formed (Figure 5 E-G). However, incomplete particles are known to be significantly less stable than complete packaged particles. This means there may be a bias in the successful isolation of packaged particles compared to incomplete particles by CsCl gradient purification. For this reason, we are reluctant to calculate and present data that would compare total capsid

assembly under conditions where the ratio of empty to packaged particles differ. In any case, we believe that the data presented robustly establish our central claim that phosphorylation promotes assembly of complete packaged particles.

With regards to further exploration into the direct functional consequences of these modifications. We agree that a characterization of 52K at the molecular level (e.g., structure/(dis)order transitions of 52K, its oligomerization etc.) could provide additional detail of value. To address oligomerization, we have now analyzed our recombinant MBP-tagged proteins (WT 52K and the S28/75A and S28/75D mutants by Native PAGE to investigate the oligomeric state of 52K. Without removal of the MBP tag, all proteins were resolved as both higher molecular weight oligomers greater than 1048 kDa, and lower molecular weight oligomers consistent with the expected size of MBP-52K dimers (~180 kDa). Removal of the MBP tag resulted in greater retention of 52K in the well of the gel, likely resulting from condensate formation. The 52K protein formed higher molecular weight oligomers that exhibited heterogenous progression through the gel, as well as lower molecular weight oligomers consistent with the expected size of 52K dimers (~94 kDa). These data indicate that 52K forms a dimer capable of higher-order oligomerization, and that this oligomerization is not dependent on its phosphorylation status. These data are included as Figure EV5 C and discussed in the result section (Lines 234-247).

In addition to the above-mentioned experiments, we have attempted several different approaches to further characterize 52K (WT and phosphorylation mutants) with regards to oligomerization and order/disorder transitions. Unfortunately, the intrinsic behavior of the 52K protein, specifically its striking propensity for dynamic oligomerization (even with MBP-tag attached), prevented meaningful interpretation. We tested Size Exclusion Chromatography with Multi Angle Light Scattering (SEC-MALS) to see if we could observe changes in the stokes radius of 52K that could provide insight into how globular vs disordered our proteins are. However, the 52K protein voided off the column, registering mostly very high molecular species indicative of high-order oligomerization. This prevented us from determining the stokes radius of 52K proteins. We also tried analytical ultracentrifugation as an orthogonal approach to investigate the hypothesis that 52K forms a dimer. However, in this instance, 52K rapidly pelleted as heterogenous high-order oligomers, again preventing meaningful interpretation of its typical lower order oligomeric state. We can provide the reviewers with reports of these experiments upon request.

2. The study shows that serine residues 28 and 75 within the N-terminal IDR of 52K are crucial for the modulation of viral condensates. However, more evidence is needed to demonstrate the specificity of these phosphorylation events given that the ADP-Glo assay was carried out with peptides. Are these the only residues that play this role, or are there other phosphorylation sites that have not been studied? Moreover, can the authors comment on the localization and activity of the identified top-ranked kinases shown in Fig EV1C - presumably, these should be active in the nucleus to phosphorylate the 52K where it forms inclusions? Lastly, if the authors know which kinases are essential for 52K phosphorylation, can they identify suitable kinase inhibitors, even low-specificity ones, which could be used to enhance their studies? Lastly, in vivo their analysis was carried out at 24 hpi (Line 157) - it is likely that these sites may dynamically change throughout the infection cycle and will vary with the multiplicity of infection. Did the authors look at the sites at earlier points of infection, and varying MOI.

Phosphorylated residues within the 52K protein were detected in samples from virus infected cells by mass spectrometry adapted for identification of post-translational modifications. Phosphorylation of S28 and S75 as well as S145 and S336 were detected. This is indicated in Figure 1A-C. We focused on phosphorylation of S28 and S75, as these sites are located within the N-terminal IDR previously shown to be critical for condensate formation and infectious progeny production (PMID: 37020020). We anticipated that phosphorylation of these sites would shed light on how the regulation of condensate properties impacts condensate function, an important question in the wider field of condensates and phase transitions. When we investigated how phosphorylation of 52K impacts condensate formation *in vitro*, we found that phosphorylation promotes the liquid-like properties of condensate fusion and surface wetting. S145 and S336 were not required for this phenotype, confirming that these liquid-like properties are conferred by phosphorylation of S28 and S75 (Figure 2C and Figure EV3B).

With regards to the localization of kinases. Yes, this is certainly interesting to think about (although we remain cautious of overstating the significance, given that we have not yet identified the kinase responsible for phosphorylation of 52K. Out of the 25 kinases screened, the majority localize to the nucleus, as reported by the human protein atlas subcellular localization study (PMID: 28495876). These include the 7 CMGC group kinases identified as top ranked hits. This is now briefly discussed in the text (lines: 136-137).

Through our bioinformatic analysis and kinase screen, we have identified kinases capable of phosphorylating the identified sites (S28 & S75). We used this information to provide some insight into which kinases may be responsible, and to help select a kinase that could phosphorylate 52K *in vitro*. However, we do not yet know which kinase or kinases are essential for phosphorylation *in vivo*. In an attempt to identify the pathways essential for phosphorylation of 52K, we have now investigated a number of different kinase inhibitors, including inhibitors of MAPK and CDK family kinases. However, we have not been able to identify an inhibitor that prevents phosphorylation of 52K (Figure EV3 A-C). This suggests that either there is redundancy, or that 52K is phosphorylated by a kinase that is not impacted by the inhibitors used. We also investigated inhibitors targeting CLK and GSK3 kinases. However, we were not able to verify efficacy of the inhibitors using available controls. For this reason, we did not include these data. To avoid any confusion, we now state clearly at the end of the result section that the responsible kinase has yet to be identified (lines: 148-151).

The 52K protein is only expressed during the late stage of infection once viral genome replication has started and the viral major late promoter is active. A time-course of infection indicates that phosphorylation of S28 and S75 is readily detected by mass spec at 16-, 24- and 48-hours post-infection. These data have now been included (Figure EV1). This would suggest that phosphorylation of these residues is maintained throughout the late stage of infection. This is now detailed in our results section (lines: 108-110). This is also consistent with the fact that permanently mimicking phosphorylation by expressing the S28/75D mutant *in trans* rescues progeny production of the 52K-null virus.

3. Can the authors provide an explanation for why the observed *in vitro* droplets coalesce better when phosphorylated (larger and fewer droplets, line 216), while in cells they observe the opposite effects (smaller and more numerous when phosphorylated, line 236).

We suspect that this is because *in vitro*, if condensates remain liquid, then condensate fusion can occur in the dish because they are free to move about and/or will contact each other when they settle. This is not the case in cells ectopically expressing 52K, since the nuclear bodies are not free to move around the

crowded nucleus. The larger nuclear bodies formed by the S28/75A mutant is interesting and mimics what we have seen previously with other mutants (PMID: 37020020). That said, we are cautious in trying to interpret the size difference between condensates observed in cells and condensates observed *in vitro*, since so many different factors might be relevant. Typically, condensates formed *in vitro* are much larger than condensates formed from the same protein in cells, and to the best of our knowledge, exactly why such large differences occur is not fully understood, although condensate fusion is likely involved. Condensate size may be influenced by multiple factors, in particular, the free-energy cost of the interface between the condensed-phase and light-phase in relation to other thermodynamic considerations including the nature of multi-valent interactions (PMID: 34315935). This might explain why modulating the interactions (e.g., phosphorylation vs no phosphorylation) that underpin condensates may result in altered condensate size in cells. Perhaps the most interesting question in this regard is why do all phase-separated condensates in cells not form a single droplet to limit the interfacial free-energy cost?

4. Fig.3 - the use of 1,6 hexanediol. Although I appreciate the usefulness of 1,6HD in such studies, I would be extremely careful in interpreting these results, particularly given the inhibitory effects of this diol on cellular kinases that are probably involved in 52K phosphorylation (PMCID: PMC7948595; 2021). The authors could try propylene glycol, which has been used in viral studies (PMCID: PMC8561643; 2021) and appear to be much better tolerated by cells over extended periods of time. Importantly, can the authors demonstrate that *in vitro* 75 mM 1,6HD does alter the phase behavior of 52K condensates (I really doubt that it does at this low concentration given multiple studies of this alcohol and how it modulates condensates). If this is the case, I doubt that the observed effect of 75 mM hexanediol has anything to do with its disruptive effect on condensates, and it has more something to do with its toxicity and inhibitory effects on cells.

We do not believe that our experiment results are due to effects of 1,6-hexanediol on cellular kinases. While we agree that 1,6-hexanediol may have both known and unknown “off target” effects (particularly at the high concentrations often used for acute disruption of condensates), any such effects should be the same for all conditions investigated (WT, S28/75A, S28/75D) which were analyzed in parallel. Importantly, if cellular kinases were inhibited under the conditions used, we would expect the opposite result to what we observed. Specifically, if cellular kinases were antagonized by 1,6-hexanediol treatment, then we would expect WT 52K (phosphorylation competent) to behave like the phosphorylation negative S28/75A mutant, while the constitutive phosphorylation mimic (S28/75D) would be phenotypically different. Clearly this is not the case, since WT mimics the S28/75D mimetic mutant, while the S28/75A mutant is phenotypically different (Figure 3F-K). In addition, we do not observe differences in the electrophoretic mobility of 52K when treated with 1,6-hexanediol (Figure 3F & I), further suggesting that 1,6-hexanediol does not inhibit phosphorylation of 52K.

Nevertheless, we agree that including experiments with propylene glycol will provide additional pertinent data. We have now repeated the 1,6-hexanediol experiments using propylene glycol. Although higher concentrations of propylene glycol are required compared to 1,6-hexanediol (likely due propylene glycol being a less aggressive disruptor), the results phenocopy those observed with 1,6-hexanediol. Specifically, the S28/75A mutant is less sensitive to antagonism by propylene glycol than WT 52K and the S28/75D mutant. These data are included (Figure EV8) and described in the text (lines: 295-303).

For additional context we have also added live/dead viability data for cells incubated with 1,6-hexanediol or propylene glycol over a range of concentrations (Figure EV7). These data demonstrate that cells are alive and membranes intact at the concentrations used for prolonged incubation. This is discussed in the results (line: 284).

With regards to the use of 1,6-hexanediol *in vitro*, this is an interesting suggestion, although concentrations of 1,6-hexanediol sufficient for antagonizing condensate formation in cells may not correspond to concentrations used *in vitro*. Phase-separation *in vitro* does not fully replicate the complex situation in cells, and so there are several reasons why the required concentrations may differ: 1) The concentration of 1,6-hexanediol required should be dependent on the amount of 52K. We are not able to determine reliably the typical concentration of 52K in cells, and so cannot match the concentrations used *in vitro* accordingly. 2) The nucleus is a crowded environment full of other factors that may theoretically interact with 52K and modify its propensity for phase-separation. This is not the case *in vitro*. 3) The conditions *in vitro* attempt to mimic approximately the conditions of the cell, but cannot reproduce exactly the solvent conditions of the nucleoplasm, all of which contribute to the balance of interactions (and entropy of the system) that affect phase-separation.

That said, if 1,6-hexanediol affects 52K in a way that is altered by phosphorylation status, we might expect that this general trend should hold true *in vitro*. We therefore tested a range of concentrations of 1,6-hexanediol, and investigated the ability of 1,6-hexanediol to attenuate condensate formation of WT, S/A, or S/D *in vitro*. We found that while the concentration required to antagonize condensate formation *in vitro* was higher than the concentration required in cells, the phosphorylation mimetic mutant was indeed more sensitive to disruption by 1,6-hexanediol than the unphosphorylated WT and S/A mutants (Figure EV9). This agrees with observations made in cells and fits well with our central conclusions. These findings are discussed in the results (lines: 295-303).

5. FRAP studies: the reported slower recovery rates - are the authors confident that these differences are directly attributed to the phosphorylation status or differences in the molecular composition of such inclusions? Do they know if lower/higher amounts of e.g., DNA in distinct puncta may also contribute to slower/faster diffusion? Do they know if phosphorylation may alter the oligomerization of their proteins, which would in turn change the diffusion and recovery rates?

Immunoprecipitation of WT and mutant 52K co-precipitates comparable levels of viral proteins (Figure 1G) and viral genomes (Figure 1H-J), suggesting that the interaction of 52K with these factors is not impacted significantly by phosphorylation of its N-terminal IDR. For this reason, we think it unlikely that altered nuclear body composition (e.g., differential recruitment of viral capsid proteins, or DNA content) is the cause of the different recovery rates observed. As mentioned above, we investigated oligomerization state by NativePAGE, but did not observe any notable difference between WT 52K, S28/75A, or S28/75D.

6. The results of virus validation studies are also intriguing: the authors show that although S28/75A mutant can contribute to infectious progeny production, this ability is attenuated compared to phosphomimetics. How comparable are the phosphomimetic and the WT viral production in this case? Are they functionally interchangeable?

Yes, both WT 52K and the phosphomimetic S28/75D mutant rescue progeny production of the Δ 52K virus to comparable levels (Figure 5B). This is interesting, as it suggests that constitutively mimicking the phosphorylated state is sufficient to promote progeny production, implying that dynamic phosphorylation of the N-terminal IDR at S28 & S75 is not required. This is consistent with our data showing that phosphorylation of these residues is maintained throughout the late stage of infection (Figure EV1). That said, even when complemented with WT 52K, the Δ 52K mutant virus produces less progeny than true WT virus infection (Figure 5B). This is something we always see, and is not surprising given that it is not possible to match exactly the natural kinetics of 52K expression when 52K is provided *in trans*.

7. Lastly, the important conclusion of the paper is the notion that 52K phosphorylation is involved in AdV genome packaging. What I find intriguing in Fig5. is that the amount of viral DNA produced when the mutant virus replicates is most likely lower compared to the WT virus (the authors should provide further data to compare DNA synthesis in all viruses and mutants examined). This is important for their interpretation, as the lower packaging efficiency could be simply attributed to lower DNA synthesis, or slower DNA replication kinetics, in which case more empty particles could be formed. I appreciate studies of genome packaging efficiencies are tricky and hard to quantify, but a lot of caution must be taken when interpreting such subtle differences in packaging, as shown in Fig.5. Additionally, the authors interpret packaging differences as an attempt to generate failed assembly intermediates enriched for 52K (Line 348) - can they also analyze their viral preparations (shown in Fig.5E) by mass-spectrometry to see if there are any differences in protein abundances?

The reviewer raises a very good point. Studying viral packaging efficiencies can be very tricky, particularly when the viral proteins required for genome-replication are also required for genome packaging, as is the case for many RNA viruses. However, in the case of adenovirus, dedicated viral packaging factors (which are expressed during the late stage of infection) are not required for DNA genome-replication. This makes adenovirus a highly tractable system to study packaging. We previously demonstrated that 52K is dispensable for genome replication (PMID: 37020020). We have now reproduced this finding, demonstrating that both WT AdV and the Δ 52K mutant replicate their genomes effectively as determined by qPCR (Figure EV11A). This important point is now clearly communicated in the text (lines: 335-339).

In addition, we have also compared genome replication of the Δ 52K mutant virus in cells expressing WT 52K, S28/75A, or S28/75D to ascertain whether differences in the number of infectious/packaged progeny particles produced might result from differences in the number of genomes replicated under these conditions. We find that genome replication of the Δ 52K AdV mutant is comparable under all conditions (Figure EV11B). This indicates that reduction in packaging with S28/75A is not due to a reduction in available genomes. These data are now described in the text (lines: 359-361).

Besides 52K, we do not see any striking differences in the abundance of viral protein when the viral particles are analyzed by SDS-PAGE and WB. While it is possible that viral particles produced under the different conditions may vary in ways yet unknown, we believe that exploring particle composition in such detail lies outside the scope of this current manuscript.

Minor:

1. Introduction: The authors may consider including more references on the role and functional consequences of scaffold phosphorylation in the formation of viral inclusions, and genome replication and packaging.

As suggested by the reviewer, we have now included additional references (PMCID: PMC6912106 & PMC8561643). We have also expanded our discussion to address similarities and differences in the role of phosphorylation with regards to viral compartments and packaging (lines: 450-477).

2. Figure 1. AlphaFold predictions - given that AlphaFold is not very good at assessing something which is supposed to lack the structure (IDPs), I suggest expressing the regions of interest and the 52K and checking their secondary structure content by circular dichroism spectroscopy. Similar experiments with phosphomimetics would also show if there are any structural rearrangements or ordering of IDRs upon phosphorylation.

This is a very valid point and a good idea. We agree that it is possible that phosphorylation could result in subtle rearrangements in secondary structure that could impact the nature of dynamic interactions. However, we anticipate that interpretation of circular dichroism spectroscopy data could be complicated with the 52K protein system. We purify 52K as a maltose binding protein (MBP) tagged fusion protein. This tag can be cleaved off to induce phase-separation but must be retained for characterization. Since MBP is large and structured, it will contribute significantly to the circular dichroism spectroscopy data. This may make it hard to reliably identify disorder to order transitions amongst the noise of MBP and structured regions of 52K. After consultation with our biophysics core director (who has significant experience in this area), we were advised against this line of investigation.

We attempted to express and purify a fusion of MBP and only the N-terminal IDR of 52K, to ask a slightly different question that might be easier to analyze. However, despite sequence validation of expression constructs, we could not get this MBP-IDR fusion protein to express in *E. coli* (incidentally, it also does not express in mammalian cells). As an alternate approach, we tried Size Exclusion Chromatography with Multi Angle Light Scattering (SEC-MALS) to see if we could observe changes in the stokes radius of 52K that could provide insight into how globular vs disordered our proteins are. However, as mentioned earlier in our response, this approach also failed to generate meaningful data, in this case due to the behavior of the protein through the column.

However, we appreciate the suggestion, and have not yet given up on finding a creative solution in the future.

3. Lines 262&265: can the authors provide errors for the reported percentage numbers of cells analyzed?

Yes, we have now provided standard deviation in the text.

4. Same for reported FRAP recovery rates, can the authors provide the errors, please?

Yes, also provided as standard deviation in the text.

Referee #2:

The author previously reported that human Adenovirus (AdV) virus assembly takes place in phase separated compartments in the nuclei of infected cells, termed biomolecular condensates (BMCs). The assembly of infectious AdV requires a number of characterized viral genes products, including the L1-52K protein investigated in this study. These virus assembly proteins interact with a specific packaging domain in the viral genome and/or with each other and virus capsid proteins.

In this report, the authors further characterize the role of the AdV L1-52K protein previously shown to nucleate the formation of BMCs. The L1-52K protein was previously shown to be phosphorylated at several specific sites. The authors validate these findings and show that two sites of phosphorylation are located in the disordered N-terminal region of the protein. They make and analyze two mutant proteins: one changes the two phosphorylated serine residues to alanines, and the second replaces the serines with aspartic acid residues as phospho-mimetics. In a convincing series of experiments, the authors show that phosphorylation of serine residues 28 and 75 within the N-terminal disordered region of L1-52K protein regulates viral condensates in vitro and in vivo. The phospho-mimetic mutant functions like wild-type, whereas the alanine substitution mutant displays altered properties of solubility and mobility in vivo. These phenotypes parallel results examining virus assembly and infectious virus production. The authors conclude that L1-52K phosphorylation alters the physical properties of the protein to promote proper BMC assembly and recruitment of AdV virus assembly and DNA packaging machinery.

This a thorough study with well designed experiments and convincing results. The manuscript is written for a broad audience. The findings are a bit specialized and perhaps of greater interest to virologists than the general readership. But the results are important and extend to a significant extent the author's recent report about phase separated AdV assembly compartments in the nucleus.

I have no major concerns about the manuscript. The authors suggest that L1-52K protein phosphorylation may stabilize multivalent interactions of this protein. Do the authors have any in vitro biophysical data to support this idea? This would greatly strengthen the manuscript, although the studies as presented stand alone of their own merit.

As the reviewer may well know, characterizing the dynamic, multi-valent interactions that underpin condensate formation is exceptionally challenging. Particularly if these interactions are highly dynamic, as is likely the case here. We did however try several different approaches aimed at characterizing oligomerization of 52K and order/disorder transitions – which could provide some insight into how phosphorylation results in multivalent interactions. Unfortunately, the intrinsic propensity of 52K to oligomerize dynamically prevented meaningful interpretation of both Size Exclusion Chromatography with Multi Angle Light Scattering (SEC-MALS) and analytical ultracentrifugation data (see also response to reviewer 1, major point 1). While we are excited by the challenge of finding creative experimental approaches to characterize these multivalent interactions better, this is something that will require further dedicated exploration.

Minor points:

1. The standard nomenclature in the AdV field is to indicate the coding region along with the molecular weight. For clarity, the authors should call the 52K protein L1-52K. Similarly call 100K L4-100K, etc. This should be done to avoid any confusion (e.g., E1B-55K vs. L1-52/55K, called 52K here).

We thank the reviewer for this clarification. We have now modified the text in line with this convention. Specifically, at first mention in the main text, we now use the terminology L1-52/55K (line: 73) and L4-100K (line: 382). Other viral proteins mentioned in the manuscript already used this convention (E1B-55K) or are not known by their molecular weight. We believe that clearly defining L1-52/55K at first mention using the standard nomenclature should be more than enough to avoid confusion between 52K and E1B-55K, the latter named in full throughout. We have also defined L1-52/55K in full in our closing paragraph (line: 479). We believe the inclusion of the full standard nomenclature at both the beginning and end of our manuscript will aid the reader in interpreting our findings in conjunction with the existing literature. Out of convenience and brevity, we have continued to use the abbreviated name for these proteins (52K and 100K respectively) elsewhere.

2. Fig. 1J. Why is the DBP coding region enriched in the ChIP analysis? This does not make sense since the packaging domain is at the left end of the viral genome.

This is an interesting point, and one we have considered ourselves. Based on the literature, one might expect enrichment of the packaging sequence specifically. However, on reflection, our data may make more sense than it first seems. The ChIP protocol does not isolate specific regions/genes, it pulls down fragments of DNA that are probed for the presence of specific regions/genes. Viral genomes are relatively small, and thus, might undergo limited fragmentation. It is possible therefore, that genome fragments pulled down via an interaction with the packaging sequence could also contain the DBP coding region which is positioned relatively close to the left end of the genome. Alternatively, it is possible that contrary to dogma, the interaction of 52K with DNA does not require a specific DNA sequence. Some preliminary evidence we have gathered (not shown here or in the manuscript) suggests that 52K can interact with plasmid DNA that does not contain a packaging sequence. Clearly, the packaging sequence is critical for encapsidation during infection, but this requirement may not be directly related to the interaction of 52K with DNA and might instead be related to the function of other packaging proteins reported to bind at the packaging sequence (e.g., IVa2). We have decided to report our findings and leave such speculation up to the reader. That said, we believe that the reviewer raises a good question worthy of thorough investigation in the future.

3. Lines 331-332: mature AdV bands at 1.34 g/cc and empty particles band at 1.29 g/cc (not 1.43 and 1.34 as stated).

Yes, thanks for catching this. We have corrected.

Prof. Matthew D Weitzman
The Children's Hospital of Philadelphia
Pathology & Laboratory Medicine
4050 Colket Translational Research Building
3501 Civic Center Blvd
Philadelphia, PA 19104

4th Dec 2023

Re: EMBOJ-2023-115032R
Phosphorylation regulates viral biomolecular condensates to promote infectious progeny production

Dear Matthew,

Thank you for submitting your revised manuscript to The EMBO Journal. It has now been seen once more by the two original referees, and I am happy to say that both were generally satisfied with your revisions and responses to the initial comments (see comments below). We are therefore ready to accept the study for publication, following incorporation of the following few editorial points:

- On the abstract page of the manuscript, please include 4-5 general keyword terms to enhance searchability.
- In the reference list, please list the individual articles only with their formal citation but not with DOI links (please refer to our Guide to Authors for additional information on EMBO J reference format). It may also be appropriate to replace the reference to a preprint by Lu et al 2020 with a reference to the peer-reviewed version in Nat Comms 2021?
- Please rename the Conflict of Interest section into "Disclosure and Competing Interests Statement", in accordance with our update Guide to Authors (<https://www.embopress.org/competing-interests>)
- Please rename the Availability of data... section into "Data Availability", remove the reviewer access notes for the deposited data, and instead add a direct link (i.e. PRIDE website combined with access identifier) to them. Also ensure that these data are now ready to be released concomitantly with acceptance.
- For the EV movies, I could not easily find the respective legends. As explained in our Guide to Authors, each movie file should be combined with an individual text file containing its respective legend into a separate ZIP archive, before re-uploading each archive as "Movie EV1/2/3..."
- Finally, I noticed that the article currently features only 5 main figures but 11 Expanded View figures, however EV figures are usually limited to 5 (max 6-7) per article. I would therefore encourage you to check if some of the EV data could be promoted to main figures (of which we could easily have 7-8), and others, less important ones "relegated" to Appendix Figures, constituting our 3rd level of data. In this case, these should be moved, together with their respective legends, into a single PDF called Appendix and prefaced by a Table of Contents listing these figures & legends. In either case, please make sure to double-check proper renumbering of main, EV and any Appendix figures (citation for the latter: "Appendix Figure S1/2/3..."). For more details about the figure levels, please refer to our guide to authors (www.embopress.org/page/journal/14602075/authorguide#expandedview)

I am therefore returning the manuscript to you for a final round of minor revision, to allow you to make these adjustments and upload all modified files. Once we will have received them, we should be ready to swiftly proceed with formal acceptance and production of the manuscript.

With kind regards,

Hartmut

*** PLEASE NOTE: All revised manuscript are subject to initial checks for completeness and adherence to our formatting guidelines. Revisions may be returned to the authors and delayed in their editorial re-evaluation if they fail to comply to the

following requirements (see also our Guide to Authors for further information):

9) Digital image enhancement is acceptable practice, as long as it accurately represents the original data and conforms to community standards. If a figure has been subjected to significant electronic manipulation, this must be clearly noted in the figure legend and/or the 'Materials and Methods' section. The editors reserve the right to request original versions of figures and the original images that were used to assemble the figure. Finally, we generally encourage uploading of numerical as well as gel/blot image source data; for details see: embopress.org/page/journal/14602075/authorguide#sourcedata

At EMBO Press, we ask authors to provide source data for the main manuscript figures. Our source data coordinator will contact you to discuss which figure panels we would need source data for and will also provide you with helpful tips on how to upload and organize the files.

In the interest of ensuring the conceptual advance provided by the work, we recommend submitting a revision within 3 months (3rd Mar 2024). Please discuss the revision progress ahead of this time with the editor if you require more time to complete the revisions. Use the link below to submit your revision:

Link Not Available

Referee #1:

I believe the authors have addressed most of the points raised during the first round of the review process. Some questions

remain open, as the authors point out in their written response; however, overall the latest version of the manuscript has been improved.

Referee #2:

The human adenovirus (AdV) genome is packaged into virus particles using viral cis-acting DNA packaging sequences and viral trans-acting proteins that interact with the packaging region, including the L1-52/55K protein (52K) studied here. The authors previously reported that AdV assembly takes place in phase separated compartments in the nuclei of infected cells, termed biomolecular condensates (BMCs). In this report, the authors further characterize the function of the 52K protein previously shown to nucleate the formation of BMCs. The 52K protein was previously shown to be phosphorylated at several specific sites. The authors validate these findings and show that two sites of phosphorylation are located in the disordered N-terminal region of the protein. They make and analyze two mutant proteins: one changes the two phosphorylated serine residues to alanines, and the second replaces the serines with aspartic acid residues as phospho-mimetics. In a convincing series of experiments, the authors show that phosphorylation of serine residues 28 and 75 within the N-terminal disordered region of 52K protein regulates viral condensates in vitro and in vivo. The phospho-mimetic mutant functions like wild-type, whereas the alanine substitution mutant displays altered properties of solubility and mobility in vivo. These phenotypes parallel results examining virus assembly and infectious virus production. The authors conclude that 52K phosphorylation alters the physical properties of the protein to promote proper BMC assembly and recruitment of AdV virus assembly and DNA packaging machinery.

This is a thorough study with well designed experiments and convincing results. The manuscript is written for a broad audience. The results are important and extend to a significant extent the author's recent report about phase separated AdV assembly compartments in the nucleus. The authors responded appropriately to the comments in my prior review and made relevant revisions.

All editorial and formatting issues were resolved by the authors.

Prof. Matthew D Weitzman
The Children's Hospital of Philadelphia
Pathology & Laboratory Medicine
4050 Colket Translational Research Building
3501 Civic Center Blvd
Philadelphia, PA 19104

7th Dec 2023

Re: EMBOJ-2023-115032R1
Phosphorylation regulates viral biomolecular condensates to promote infectious progeny production

Dear Prof. Weitzman,

Thank you for submitting your final revised manuscript for our consideration. I am pleased to inform you that we have now accepted it for publication in The EMBO Journal.

Yours sincerely,

Hartmut Vodermaier
